# Continuous Product Graph Neural Networks

**Aref Einizade**
LTCI, Télécom Paris
Institut Polytechnique de Paris
`aref.einizade@telecom-paris.fr`

**Fragkiskos D. Malliaros**
CentraleSupélec, Inria
Université Paris-Saclay
`fragkiskos.malliaros@centralesupelec.fr`

**Jhony H. Giraldo**
LTCI, Télécom Paris
Institut Polytechnique de Paris
`jhony.giraldo@telecom-paris.fr`

## Abstract

Processing multidomain data defined on multiple graphs holds significant potential in various practical applications in computer science. However, current methods are mostly limited to discrete graph filtering operations. Tensorial partial differential equations on graphs (TPDEGs) provide a principled framework for modeling structured data across multiple interacting graphs, addressing the limitations of the existing discrete methodologies. In this paper, we introduce **Con**tinuous Produc**t** G**r**aph Ne**u**ral Network**s** (CITRUS) that emerge as a natural solution to the TPDEG. CITRUS leverages the separability of continuous heat kernels from Cartesian graph products to efficiently implement graph spectral decomposition. We conduct thorough theoretical analyses of the stability and over-smoothing properties of CITRUS in response to domain-specific graph perturbations and graph spectra effects on the performance. We evaluate CITRUS on well-known traffic and weather spatiotemporal forecasting datasets, demonstrating superior performance over existing approaches. The implementation codes are available at https://github.com/ArefEinizade2/CITRUS.

## 1 Introduction

Multidomain (tensorial) data defined on multiple interacting graphs [1–3], referred to as multidomain graph data in this paper, extend the traditional graph machine learning paradigm, which typically deals with single graphs [2, 4]. Tensors, which are multi-dimensional generalizations of matrices (order-2 tensors), appear in various fields like hyperspectral image processing [5], video processing [6], recommendation systems [7], spatiotemporal analysis [8], and brain signal processing [9]. Despite the importance of these applications, learning from multidomain graph data has received little attention in the existing literature [2, 10]. Therefore, developing graph-learning strategies for these tensorial data structures holds significant promise for various practical applications.

The main challenge for learning from multidomain graph data is creating efficient frameworks that model *joint* interactions across domain-specific graphs [10, 11]. Previous work in this area has utilized discrete graph filtering operations in product graphs (PGs) [10, 12] from the field of graph signal processing (GSP) [13]. However, these methods inherit the well-known issues of over-smoothing and over-squashing from regular graph neural networks (GNNs) [14–16], which restricts the graph's receptive field and hinders long-range interactions [17]. Additionally, these methods often require computationally intensive grid searches to tune hyperparameters and are typically limited to two-domain graph data, such as spatial and temporal dimensions [10, 12, 18, 19].

38th Conference on Neural Information Processing Systems (NeurIPS 2024).

In regular non-tensorial GNNs, continuous GNNs (CGNNs) emerge as a solution to over-smoothing and over-squashing [20]. CGNNs are neural network solutions to partial differential equations (PDEs) on graphs [20, 21], like the heat or wave equation. The solution to these PDEs introduces the exponential graph filter as a continuous infinity-order generalization of the typical discrete graph convolutional filters [20]. Due to the differentiability of exponential graph filters w.r.t. the graph receptive fields, CGNNs can benefit from both global and local message passing by adaptively learning graph neighborhoods, alleviating the limitations of discrete GNNs [17, 20, 22]. Despite these advantages, CGNNs mostly rely on a *single* graph and lack a principled framework for learning *joint* multi-graph interactions.

In this paper, we introduce tensorial PDE on graphs (TPDEGs) to address the limitations of existing PG-based GNN and CGNN frameworks. TPDEGs provide a principled framework for modeling multidomain data residing on multiple interacting graphs that form a Cartesian product graph heat kernel. We then propose **C**ontinuous Produc**t** G**r**aph Ne**u**ral Network**s** (CITRUS) as a continuous solution to TPDEGs. We efficiently implement CITRUS using a small subset of eigenvalue decompositions (EVDs) from the factor graphs. Additionally, we conduct theoretical and experimental analyses of the stability and over-smoothing properties of CITRUS. We evaluate our proposed model on spatiotemporal forecasting tasks, demonstrating that CITRUS achieves state-of-the-art performance compared to previous Temporal-Then-Spatial (TTS), Spatial-Then-Temporal (STT), and Temporal-and-Spatial (T&S) frameworks. Our main contributions are summarized as follows:

- We introduce TPDEGs to handle multidomain graph data. This leads to our proposal of a continuous graph filtering operation in Cartesian product spaces, called CITRUS, which naturally emerges as the solution to TPDEGs.

- We conduct extensive theoretical and experimental analyses to evaluate the stability and over-smoothing properties of CITRUS.

- We test CITRUS on spatiotemporal forecasting tasks, demonstrating that our model achieves state-of-the-art performance.

## 2   Related Work

We divide the related work into two parts covering the *fundamental* and *applicability* aspects.

**Fundamentals.** The study of PGs is a well-established field in mathematics and signal processing [23, 24]. However, to the best of our knowledge, the graph-time convolutional neural network (GTCNN) [10, 18] model is the only GNN-based framework addressing neural networks in PGs. GTCNN defines a discrete filtering operation in PGs to jointly process spatial and temporal data for forecasting applications. Due to the discrete nature of the graph polynomial filters in GTCNN, an expensive grid search is required to correctly tune the graph filter orders, leading to significant computational overhead. Additionally, this discrete aspect restricts node-wise receptive fields, making long-range communication challenging [17, 20]. Furthermore, GTCNN is limited to two factor graphs (space and time), limiting its applicability to general multidomain graph data. In contrast, CITRUS overcomes these limitations by i) employing continuous graph filtering, and ii) providing a general framework for any number of factor graphs.

**Applicability.** We explore spatiotemporal analysis as a specific application of PG processing. Early works in GSP for spatiotemporal forecasting, such as the graph polynomial vector auto-regressive moving average (GP-VARMA) and GP-VAR models [25], do not utilize PGs. Modern GNN-based architectures can be broadly classified into TTS and STT frameworks. TTS networks, due to their sequential nature, often encounter propagation information bottlenecks during training [8]. Conversely, STT architectures enhance node representations first and then integrate temporal information, but the use of recurrent neural networks (RNNs) in the aggregation stages leads to high computational complexity. Additionally, there are T&S frameworks designed for *simultaneous* learning of spatiotemporal representations [8, 26], where the fusion of temporal and spatial modules is crucial. However, similar to STT networks, T&S frameworks also face challenges with computational complexity [8] and their discretized block-wise nature, which can limit adaptive graph receptive fields [20, 26].

CITRUS leverages *joint* multidomain learning by exploiting continuous separable heat kernels as factor-based functions defined on a Cartesian product graph. The continuity and differentiability

Factor Graphs  Product Graph
$(\mathbf{L}_1 \oplus \mathbf{L}_2 \oplus \mathbf{L}_3)$

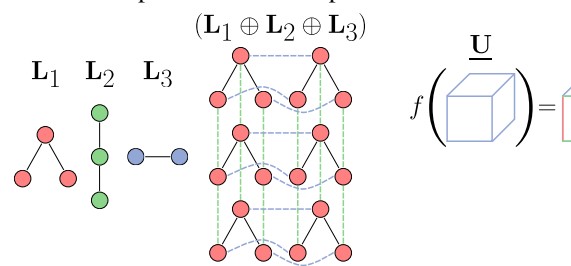

Continuous Product Graph Function (CITRUS)

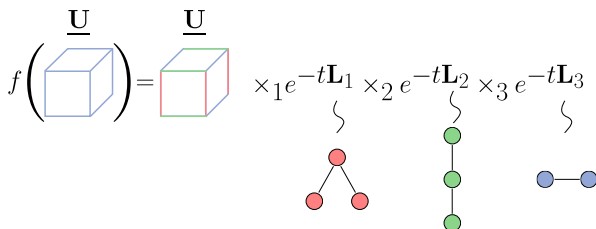

(a) Cartesian product graph.

(b) Continous product graph function in (5).

Figure 1: Illustration of key concepts of CITRUS. a) Cartesian product between three-factor graphs. b) Continous product graph function (CITRUS) operating on the multidomain graph data $\underline{\mathbf{U}}$.

of these filters allow the graph receptive fields to be learned adaptively during the training process, eliminating the need for a grid search. Additionally, CITRUS maintains low computational complexity by relying on the spectral decomposition of the factor graphs [17, 22]. Furthermore, unlike non-graph approaches, the number of learnable parameters in CITRUS is independent of the factor graphs, ensuring scalability.

## 3  Methodology and Theoretical Analysis

**Preliminaries and notation.** An undirected and weighted graph $\mathcal{G}$ with $N$ vertices can be stated as $\mathcal{G} = \{\mathcal{V}, \mathcal{E}, \mathbf{A}\}$. $\mathcal{V}$ and $\mathcal{E}$ denote the sets of the vertices and edges, respectively. $\mathbf{A} \in \mathbb{R}^{N \times N}$ is the adjacency matrix with $\{\mathbf{A}_{ij} = \mathbf{A}_{ji} \geq 0\}_{i,j=1}^{N}$ representing the connection between vertices, and $\{\mathbf{A}_{ii} = 0\}_{i=1}^{N}$ states that there are no self-loops in $\mathcal{G}$. The graph Laplacian $\mathbf{L} \in \mathbb{R}^{N \times N}$ is defined as $\mathbf{L} = \mathbf{D} - \mathbf{A}$, where $\mathbf{D} = \mathrm{diag}(\mathbf{A}\mathbf{1})$ is the diagonal degree matrix and $\mathbf{1}$ is the all-one vector. A graph signal is a function that maps a set of nodes to the real values $x : \mathcal{V} \to \mathbb{R}$, and therefore we can represent it as $\mathbf{x} = [x_1, \ldots, x_N]^\top$. For a vector $\mathbf{a} = [a_1, \ldots, a_N]^\top \in \mathbb{R}^{N \times 1}$, $e^{\mathbf{a}} := [e^{a_1}, \ldots, e^{a_N}]^\top$. The vectorization operator is shown as $\mathrm{vec}(.)$. Using the Kronecker product $\otimes$, Kronecker sum (aka Cartesian product) $\oplus$, and the Laplacian factors $\{\mathbf{L}_p \in \mathbb{R}^{N_p \times N_p}\}_{p=1}^{P}$, we have the following definitions:

$$\downarrow \oplus_{p=1}^{P} \mathbf{L}_p := \mathbf{L}_P \oplus \ldots \oplus \mathbf{L}_1, \ \ \otimes_{p=1}^{P} \mathbf{L}_p := \mathbf{L}_1 \otimes \ldots \otimes \mathbf{L}_P, \ \ \downarrow \otimes_{p=1}^{P} \mathbf{L}_p := \mathbf{L}_P \otimes \ldots \otimes \mathbf{L}_1, \quad (1)$$

where the Laplacian matrix of the Cartesian product between two factor graphs can be stated as:

$$\oplus_{p=1}^{2} \mathbf{L}_p = \mathbf{L}_1 \oplus \mathbf{L}_2 = \mathbf{L}_1 \otimes \mathbf{I}_{N_2} + \mathbf{I}_{N_1} \otimes \mathbf{L}_2, \quad (2)$$

with $\mathbf{I}_n$ the identity matrix of size $n$. See Figure 1a for an example of a Cartesian graph product between three factor graphs. A similar relationship to (2) also holds for adjacency matrices [24, 27]. We define a $D$-dimensional tensor as $\underline{\mathbf{U}} \in \mathbb{R}^{N_1 \times \ldots \times N_D}$. The matrix $\underline{\mathbf{U}}_{(i)} \in \mathbb{R}^{N_i \times [\prod_{r=1, r \neq i}^{D} N_r]}$ is the $i$-th mode matricization of $\underline{\mathbf{U}}$ [28]. The mode-$i$ tensorial multiplication of a tensor $\underline{\mathbf{U}}$ by a matrix $\mathbf{X} \in \mathbb{R}^{m \times N_i}$ is denoted by $\underline{\mathbf{G}} = \underline{\mathbf{U}} \times_i \mathbf{X}$, where $\underline{\mathbf{G}} \in \mathbb{R}^{N_1 \times \ldots \times N_{i-1} \times m \times N_{i+1} \times \ldots \times N_D}$ [28]. For further information about the tensor matricization and mode-$n$ product, please refer to [28, 29] (especially Sections 2.4 and 2.5 in [28]). All the proofs of theorems, propositions, and lemmas of this paper are provided in the Appendix A.

### 3.1  Continuous Product Graph Neural Networks (CITRUS)

We state the generative PDE for CITRUS as follows:

**Definition 3.1** (Tensorial PDE on graphs (TPDEG)). Let $\tilde{\underline{\mathbf{U}}}_t \in \mathbb{R}^{N_1 \times N_2 \times \ldots \times N_P}$ be a multidomain tensor whose elements are functions of time $t$, and let $\{\mathbf{L}_p \in \mathbb{R}^{N_p \times N_p}\}_{p=1}^{P}$ be the domain-specific factor Laplacians. We can define the TPDEG as follows:

$$\frac{\partial \tilde{\underline{\mathbf{U}}}_t}{\partial t} = -\sum_{p=1}^{P} \tilde{\underline{\mathbf{U}}}_t \times_p \mathbf{L}_p. \quad (3)$$

**Theorem 3.2.** *Let $\underline{\tilde{\mathbf{U}}}_0$ be the initial value of $\underline{\tilde{\mathbf{U}}}_t$. The solution to the TPDEG in (3) is given by:*

$$\underline{\tilde{\mathbf{U}}}_t = \underline{\tilde{\mathbf{U}}}_0 \times_1 e^{-t\mathbf{L}_1} \times_2 e^{-t\mathbf{L}_2} \times_3 \ldots \times_P e^{-t\mathbf{L}_P}. \tag{4}$$

Using Theorem 3.2, we define the core function of CITRUS as follows:

$$f(\underline{\mathbf{U}}_l) = \underline{\mathbf{U}}_l \times_1 e^{-t_l\mathbf{L}_1} \times_2 \ldots \times_P e^{-t_l\mathbf{L}_P} \times_{P+1} \mathbf{W}_l^\top, \tag{5}$$

where $\underline{\mathbf{U}}_l \in \mathbb{R}^{N_1 \times \ldots \times N_P \times F_l}$ is the input tensor, $\mathbf{W}_l \in \mathbb{R}^{F_l \times F_{l+1}}$ is a matrix of learnable parameters, $t_l$ is the learnable graph receptive field, and $f(\underline{\mathbf{U}}_l) \in \mathbb{R}^{N_1 \times \ldots \times N_P \times F_{l+1}}$ is the output tensor. Figure 1b illustrates a high-level description of $f(\underline{\mathbf{U}}_l)$. The next proposition formulates (5) as a graph convolution defined on a product graph.

**Proposition 3.3.** *The core function of CITRUS in* (5) *can be rewritten as:*

$$\left[f(\underline{\mathbf{U}}_l)_{(P+1)}\right]^\top = e^{-t_l\mathbf{L}_\diamond}[\underline{\mathbf{U}}_{l(P+1)}]^\top \mathbf{W}_l, \tag{6}$$

*where $\mathbf{L}_\diamond := \downarrow \oplus_{p=1}^{P} \mathbf{L}_p$ is the Laplacian of the Cartesian product graph.*

Proposition 3.3 is the main building block for implementing CITRUS. More precisely, we use the spectral decompositions of the factor graphs $\{\mathbf{L}_p = \mathbf{V}_p \boldsymbol{\Lambda}_p \mathbf{V}_p^\top\}_{p=1}^{P}$ and product graph $\{\mathbf{L}_\diamond = \mathbf{V}_\diamond \boldsymbol{\Lambda}_\diamond \mathbf{V}_\diamond^\top\}$, where $\mathbf{V}_\diamond = \downarrow \otimes_{p=1}^{P} \mathbf{V}_p$ and $\boldsymbol{\Lambda}_\diamond = \downarrow \oplus_{p=1}^{P} \boldsymbol{\Lambda}_p$ [23]. Let $K_p \leq N_p$ be the number of selected eigenvalue-eigenvector pairs of the $p$-th factor Laplacian. When $K_p = N_p$, it can be shown [17, 22] that we can rewrite (6) as follows:

$$\left[f(\underline{\mathbf{U}}_l)_{(P+1)}\right]^\top = \mathbf{V}_\diamond^{(K_p)} \left( \underbrace{\overbrace{[\tilde{\boldsymbol{\lambda}}_l | \ldots | \tilde{\boldsymbol{\lambda}}_l]}^{F_l \text{ times}}}_{\tilde{\boldsymbol{\Lambda}}_l} \odot \left( \mathbf{V}_\diamond^{(K_p)^\top} \left[\underline{\mathbf{U}}_{l(P+1)}\right]^\top \right) \right) \mathbf{W}_l, \tag{7}$$

$$\text{with} \qquad \tilde{\boldsymbol{\lambda}}_l = \downarrow \otimes_{p=1}^{P} e^{-t_l \boldsymbol{\lambda}_p^{(K_p)}}, \qquad \mathbf{V}_\diamond^{(K_p)} = \downarrow \otimes_{p=1}^{P} \mathbf{V}_p^{(K_p)}, \tag{8}$$

where $\boldsymbol{\lambda}_p^{(K_p)} \in \mathbb{R}^{K_p \times 1}$ and $\mathbf{V}_p^{(K_p)} \in \mathbb{R}^{N_p \times K_p}$ are the first $K_p \leq N_p$ selected eigenvalues and eigenvectors of $\mathbf{L}_p$ based on largest eigenvalue magnitudes, respectively, and $\odot$ is the point-wise multiplication operation. Finally, we can define the output of the $l$-th layer of CITRUS as $\underline{\mathbf{U}}_{l+1(P+1)} = \sigma\left(f(\underline{\mathbf{U}}_l)_{(P+1)}\right)$, where $\sigma(\cdot)$ is some proper activation function.

*Remark* 3.4. We can consider factor-specific learnable graph receptive fields $\{t_l^{(p)}\}_{p=1}^{P}$ for formulating $\tilde{\boldsymbol{\lambda}}_l$ in (8) as $\tilde{\boldsymbol{\lambda}}_l = \downarrow \otimes_{p=1}^{P} e^{-t_l^{(p)} \boldsymbol{\lambda}_p^{(K_p)}}$ to make CITRUS more flexible for each factor graph's receptive field. This technique can be expanded to the channel-wise graph receptive fields $\{t_l^{(p,c)}\}_{p=1,c=1}^{P,F_l}$ such that the $c$-th column of $\tilde{\boldsymbol{\Lambda}}_l$ in (7) is obtained by $\tilde{\boldsymbol{\lambda}}_{k,c} = \downarrow \otimes_{p=1}^{P} e^{-t_l^{(p,c)} \boldsymbol{\lambda}_p^{(K_p)}}$.

*Remark* 3.5. Computing the EVD in the product graph $\mathbf{L}_\diamond$ in (6) has a general complexity of $\mathcal{O}([\prod_{p=1}^{P} N_p]^3)$. However, we obtain a significant reduction in complexity of $\mathcal{O}([\sum_{p=1}^{P} N_p^3])$ in (7) by relying upon the properties of product graphs since we perform EVD on each factor graph independently. Besides, if we rely only on the $K_p$ most important eigenvector-eigenvalue pairs [17] of each factor graph, we can reduce the complexity of the spectral decomposition to $\mathcal{O}(N_p^2 K_p)$, obtaining a general complexity of $\mathcal{O}([\sum_{p=1}^{P} N_p^2 K_p])$.

### 3.2 Stability Analysis

We study the stability of CITRUS against possible perturbations in the factor graphs. First, as defined in the literature [30], we model the perturbation to the $p$-th graph as an addition of an error matrix $\mathbf{E}_p$ (with upper bound $\varepsilon_p$ for any matrix norm $\|\!|\cdot|\!\|$) to the adjacency matrix $\mathbf{A}_p$ as:

$$\tilde{\mathbf{A}}_p = \mathbf{A}_p + \mathbf{E}_p; \quad \|\!|\mathbf{E}_p|\!\| \leq \varepsilon_p. \tag{9}$$

**Proposition 3.6.** *Let $\mathbf{A}$ and $\tilde{\mathbf{A}}$ be the true and perturbed adjacency matrix of Cartesian product graphs with the perturbation model for each factor graph described as in* (9). *Then, it holds that:*

$$\tilde{\mathbf{A}} = \mathbf{A} + \mathbf{E}; \quad \|\!|\mathbf{E}|\!\| \leq \sum_{p=1}^{P} \varepsilon_p, \tag{10}$$

*where the perturbation matrix $\mathbf{E}$ also follows the Cartesian structure $\mathbf{E} = \oplus_{p=1}^{P} \mathbf{E}_p$.*

Finally, let $\varphi(u,t)$ and $\tilde{\varphi}(u,t)$ be the true and perturbed output of the CITRUS model with normalized Laplacian $\hat{\mathbf{L}}_\diamond$ being responsible for generating $\underline{\mathbf{U}}_{t(P+1)}$ in a heat flow PDE with normalized Laplacian $\hat{\mathbf{L}}_\diamond$ as $\frac{\partial \varphi(u,t)}{\partial t} = -\hat{\mathbf{L}}_\diamond \varphi(u,t)$ in (6) [30], respectively. Then, the following theorem states that the integrated error bound on the stability properties of CITRUS is also separable when dealing with Cartesian product graphs:

**Theorem 3.7.** *Consider a Cartesian product graph (without isolated nodes) with the normalized true and perturbed Laplacians $\hat{\mathbf{L}}_\diamond$ and $\tilde{\hat{\mathbf{L}}}_\diamond$ in (6). Also, assume we have the perturbation model $\tilde{\mathbf{A}}_p = \mathbf{A}_p + \mathbf{E}_p$ with factor error bounds $\{\|\|\mathbf{E}_p\|\| \leq \varepsilon_p\}_{p=1}^P$ in Proposition 3.6. Then, the stability bound on the true and perturbed outputs of CITRUS , i.e., $\varphi(u,t)$ and $\tilde{\varphi}(u,t)$, respectively, can be described by the summation of the factor stability bounds as:*

$$\|\varphi(u,t) - \tilde{\varphi}(u,t)\| = \sum_{p=1}^P \mathcal{O}(\varepsilon_p). \tag{11}$$

Theorem 3.7 states that the solution of the TPDEG (4) is robust w.r.t. the scale of the factor graph perturbations. This is a desired property when dealing with erroneous factor graphs, and will be numerically validated in Section 4.1.

### 3.3 Over-smoothing Analysis

There is a prevalent notion of describing over-smoothing in GNNs based on decaying Dirichlet energy against increasing the number of layers. This has been theoretically analyzed in related literature [20, 31, 32] and can be formulated [20] for the output of a continuous GNN. Precisely, it can be defined on the GNN's output $\mathbf{U}_t = [\mathbf{u}(t)_1, \ldots, \mathbf{u}(t)_N]^\top$, with $\mathbf{u}(t)_i$ being the feature vector of the $i$-th node with the degree $\deg_i$, as:

$$\lim_{t \to \infty} E(\mathbf{U}_t) \to 0, \tag{12}$$

$$\text{where} \quad E(\mathbf{U}) := \frac{1}{2} \sum_{(i,j) \in \mathcal{E}} \left\| \frac{\mathbf{u}_i}{\sqrt{\deg_i}} - \frac{\mathbf{u}_j}{\sqrt{\deg_j}} \right\|^2 = \text{tr}(\mathbf{U}^\top \hat{\mathbf{L}} \mathbf{U}). \tag{13}$$

Inspired by (13), we extend the definition of Dirichlet energy to the tensorial case as follows:

**Definition 3.8.** (Tensorial Dirichlet energy). By considering the normalized factor Laplacians $\{\hat{\mathbf{L}}_p\}_{p=1}^P$, we define the Tensorial Dirichlet energy for a tensor $\underline{\mathbf{U}} \in \mathbb{R}^{N_1 \times \ldots \times N_P \times F}$ as:

$$E(\underline{\mathbf{U}}) := \frac{1}{P} \sum_{f=1}^F \sum_{p=1}^P \text{tr}(\tilde{\mathbf{U}}_{f_{(p)}}^\top \hat{\mathbf{L}}_p \tilde{\mathbf{U}}_{f_{(p)}}), \tag{14}$$

where $\tilde{\mathbf{U}}_{f_{(p)}} = \underline{\mathbf{U}}[:, \ldots, :, f]_{(p)} \in \mathbb{R}^{N_p \times \prod_{i \neq p}^P N_i}$, *i.e.*, the $p$-th mode matricization of the $f$-th slice of $\underline{\mathbf{U}}$ in its $(P+1)$-th dimension.

Based on the previous research in the GSP literature [27, 33, 34], it follows that $E(\underline{\mathbf{U}})$ in (14) can be rewritten as $E(\underline{\mathbf{U}}) = \text{tr}(\tilde{\mathbf{U}}^\top \hat{\mathbf{L}} \tilde{\mathbf{U}})$, where $\tilde{\mathbf{U}} \in \mathbb{R}^{[\prod_{p=1}^P N_p] \times F}$, $\tilde{\mathbf{U}}[:, f] = \text{vec}(\underline{\mathbf{U}}[:, \ldots, :, f])$, and $\hat{\mathbf{L}} := \frac{1}{P} \oplus_{p=1}^P \hat{\mathbf{L}}_p = \oplus_{p=1}^P \left( \frac{\hat{\mathbf{L}}_p}{P} \right)$. The next lemma shows interesting and applicable properties of $\hat{\mathbf{L}}$.

**Lemma 3.9.** *Consider $P$ factor graphs with normalized adjacencies $\{\hat{\mathbf{A}}_p\}_{p=1}^P$ and Laplacians $\{\hat{\mathbf{L}}_p\}_{p=1}^P$. By constructing the product adjacency as $\hat{\mathbf{A}} := \frac{1}{P} \oplus_{p=1}^P \hat{\mathbf{A}}_p = \oplus_{p=1}^P \left( \frac{\hat{\mathbf{A}}_p}{P} \right)$, a similar Cartesian form for the product Laplacian $\hat{\mathbf{L}} := \mathbf{I} - \hat{\mathbf{A}}$ (with $\mathbf{I} \in \mathbb{R}^{[\prod_{p=1}^P N_p] \times [\prod_{p=1}^P N_p]}$ being the identity matrix) also holds as the following:*

$$\hat{\mathbf{L}} = \frac{1}{P} \oplus_{p=1}^P \hat{\mathbf{L}}_p = \oplus_{p=1}^P \left( \frac{\hat{\mathbf{L}}_p}{P} \right). \tag{15}$$

*Besides, the spectrum of $\hat{\mathbf{L}}$ is spanned across the interval of $[0, 2]$.*

We observe in (14) that the kernel separability of Cartesian heat kernels also leads to the separability of analyzing over-smoothing in different modes of the data at hand. This is useful in cases of facing factor graphs with different characteristics. Next, we formally analyze over-smoothing in CITRUS. In our case, for the $l$-th layer with $H_l$ hidden MLP layers, non-linear activations $\sigma(\cdot)$ and learnable weight matrices $\{\mathbf{W}_{lh}\}_{h=1}^{H_l}$, we define:

$$\mathbf{X}_{l+1} := f_l(\mathbf{X}_l), \quad f_l(\mathbf{X}) := \mathrm{MLP}_l(e^{-\hat{\mathbf{L}}^{(t)}}\mathbf{X}), \quad \mathrm{MLP}_l(\mathbf{X}) := \sigma(\ldots\sigma(\sigma(\mathbf{X})\mathbf{W}_{l1})\mathbf{W}_{l2}\ldots\mathbf{W}_{lH_l}), \tag{16}$$

with $\mathbf{X}_0$ being the initial node feature matrix, $f_l(\cdot)$ is a generalized form of (6), and $e^{-\hat{\mathbf{L}}^{(t)}} := \downarrow \otimes_{p=1}^{P} e^{-t^{(p)}\frac{\hat{\mathbf{L}}_p}{P}}$, where $t^{(i)}$ is the receptive field of the $i$-th factor graph. Then, the next theorem describes the over-smoothing criteria.

**Theorem 3.10.** *For the product Laplacian $\hat{\mathbf{L}} := \frac{1}{P}\oplus_{p=1}^{P}\hat{\mathbf{L}}_p$ with domain-specific receptive field $t^{(i)}$ for the $i$-th factor graph and the activations $\sigma(\cdot)$ in (16) being ReLU or Leaky ReLU, we have:*

$$E(\mathbf{X}_l) \leq e^{l\left(\ln s - \frac{2\tilde{t}\tilde{\lambda}}{P}\right)}E(\mathbf{X}_0), \tag{17}$$

*where $s := \sup_{l\in\mathbb{N}_+} s_l$ and $s_l := \prod_{h=1}^{H_l} s_{lh}$ with $s_{lh}$ being the square of maximum singular value of $\mathbf{W}_{lh}^{\top}$. Besides, $\tilde{\lambda} = \lambda^{(m)}$ and $\tilde{t} = t^{(m)}$ with $m = \arg\min_i t^{(i)}\lambda^{(i)}$, where $\lambda^{(i)}$ is the smallest non-zero eigenvalue of $\hat{\mathbf{L}}_i$.*

**Corollary 3.11.** *When $l \to \infty$, $E(\mathbf{X}_l)$ exponentially converges to 0, when:*

$$\ln s - \frac{2}{P}\tilde{t}\tilde{\lambda} < 0. \tag{18}$$

Theorem 3.10 shows that the factor graph with the smallest non-zero eigenvalue (spectral gap) multiplied by its receptive field dominates the overall over-smoothing. The less the spectral gap, the less probability the factor graph is connected [14]. So, we can focus on the (much smaller) factor graphs instead of the resulting massive product graph. These theoretical findings are experimentally validated in Section 4.2.

## 4 Experimental Results

In this section, we experimentally validate the theoretical discussions regarding the stability and over-smoothing analysis and use CITRUS in real-world spatiotemporal forecasting tasks. Similarly, we conduct ablation studies regarding CITRUS in spatiotemporal forecasting. We compare CITRUS against several state-of-the-art methods in forecasting. The results on the case of more than two factor graphs and descriptions of the state-of-the-art are provided in Appendix B and C, respectively, alongside the additional theoretical and experimental discussions in Sections D-I. The implementation codes are available at https://github.com/ArefEinizade2/CITRUS.

### 4.1 Experimental Stability Analysis

We generate a 600-node product graph consisting of two connected Erdős-Rényi (ER) factor graphs with $N_1 = 20$, $N_2 = 30$, and edge probabilities $p_{\mathrm{ER}}^{(1)} = p_{\mathrm{ER}}^{(2)} = 0.1$ for node regression. We utilize 15% of the nodes in the product graph as the test set, 15% of the remaining nodes as validation, and the rest of the nodes for training. We use a three-layer generative model to generate the outputs based on (6). The continuous parameters for the factor graphs are set as $t^{(1)} = 2$ and $t^{(2)} = 3$. The initial product graph signal $\mathbf{X}^{(0)} \in \mathbb{R}^{N \times F_0}$ is generated from the normal distribution with $F_0 = 6$. Each layer's MLP of the generation model has only one layer with the number of hidden units $\{5, 4, 2\}$. We add noise to the factor adjacency matrices with signal-to-noise ratios

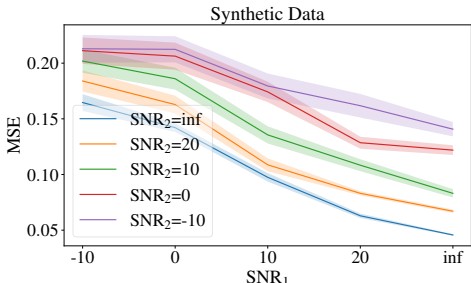

Figure 2: Stability analysis vs. different SNR scenarios, related the results in Theorem 3.7.

(SNRs) (in terms of db) of $\{\infty, 20, 10, 0, -10\}$. Therefore, we construct a two-layer CITRUS and use one-layer MLP for each with $F_1 = F_2 = 4$ to learn the representation in different SNRs to study the effect of each factor graph's noise on the stability performance of the network.

Figure 2 shows the averaged prediction error in MSE between the true and predicted outputs over 10 random realizations. Firstly, we notice that increasing the SNR for each factor graph improves performance, illustrating the stability properties of CITRUS. Therefore, the effect of each factor graph's stability on the overall stability is well depicted in this figure, confirming the theoretical findings in Theorem 3.7. For instance, in SNR=10 for the first graph, the performance is still severely affected by the stability for the second graph, shown by color in Figure 2. Finally, we observe that the standard deviation of the predictions is smaller for larger values of SNR, illustrating CITRUS's robustness.

## 4.2 Experimental Over-smoothing Analysis

For experimentally validating Theorem 3.10, we first generate a initial graph signal $\mathbf{X}_0 \in \mathbb{R}^{N \times F_0}$ with $F_0 = 12$ on a 150-node Cartesian product graph with $N_1 = 10$, $N_2 = 15$. Therefore, we consider a 10-layer generation process based on (6) with only one-layer MLP for each layer and $\{F_l = 12 - l\}_{l=1}^{10}$, where the weight matrices are generated from normal distribution. We select ReLU as the activation function and $\{t_l = 1\}_{l=1}^{10}$.

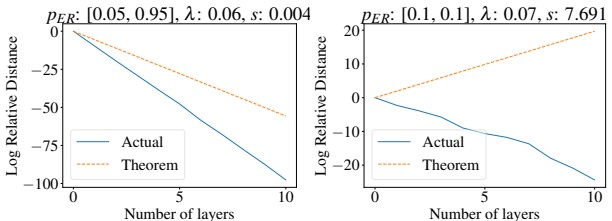

Figure 3: Over-smoothing analysis using Theorem 3.10. **left:** $\ln s - \frac{2}{P}\tilde{t}\tilde{\lambda} < 0$, **right:** $\ln s - \frac{2}{P}\tilde{t}\tilde{\lambda} > 0$.

The over-smoothing phenomenon strongly depends on the smallest spectral gap of the factor graphs as stated in Theorem 3.10. We consider two scenarios of $\ln s - \frac{2}{P}\tilde{t}\tilde{\lambda} < 0$ and $\ln s - \frac{2}{P}\tilde{t}\tilde{\lambda} > 0$. The first scenario relates to a well-connected product graph obtained from two factor ER graphs with $p_{\text{ER}}^{(1)} = 0.05$ and $p_{\text{ER}}^{(2)} = 0.95$ and, while in the other, $p_{\text{ER}}^{(1)} = p_{\text{ER}}^{(2)} = 0.1$. So, we scale the MLP weight matrices by 100 and 2.5 to limit their maximum singular values. These settings result in $\lambda = 0.06$ and $s = 0.004 \pm 10^{-5}$ for the first scenario; and $\lambda = 0.07$ and $s = 7.691 \pm 10^{-6}$ for the other.

We depict the outputs $y(l) = \log \frac{E(\mathbf{X}_l)}{E(\mathbf{X}_0)}$ and the theoretical upper bound $y(l) = l(\ln s_l - \frac{2}{P}\tilde{t}_l\tilde{\lambda})$ across the number layers $l$ in Figure 3. We observe that in the **left** plot where $\{\ln s_l - \frac{2}{P}\tilde{t}_l\tilde{\lambda} < 0\}_{l=1}^{10}$, the theoretical upper bound approximates well the actual outputs during over-smoothing as stated in Theorem 3.10. On the opposite, on the **right** plot, in which $\{\ln s_l - \frac{2}{P}\tilde{t}_l\tilde{\lambda} > 0\}_{l=1}^{10}$, the theoretical upper bound is loose. We leave for future work finding a tighter upper bound.

## 4.3 Experiments on Real-world Data

We evaluate and compare CITRUS on real-world traffic and weather spatiotemporal forecasting tasks, where we follow the settings in [10] for pre-processing and training-validation-testing data partitions. All hyperparameters were optimized on the validation set with the details in Section I. We use Adam as optimizer.

**Traffic forecasting.** Here, we work on two well-known datasets for spatiotemporal forecasting: *MetrLA* [35] and *PemsBay* [36]. *MetrLA* consists of recorded traffic data for four months on 207 highways in Los Angeles County with 5-minute resolutions [35]. *PemsBay* contains 5-minute traffic load data for six months associated with 325 stations in the Bay Area by the same setting with [36]. The spatial and temporal graphs are created by applying Gaussian kernels on the node distances [37] and simple path graphs, respectively. The task for these datasets is, by having the last 30 minutes ($T = 6$ steps) of traffic recordings, we need to predict the following 15-30-60 minutes, *i.e.*, $H = 3$, $H = 6$, and $H = 12$ steps ahead horizons. Our model first uses a linear encoder network, followed by three CITRUS blocks with 3-layer MLPs for each, where $\{F_l^{\text{MLP}} = 64\}_{l=1}^3$. Therefore, the output node embeddings are concatenated with the initial spatiotemporal data. Finally, we use a linear decoding layer to transform the learned representations to the number of needed horizons. We exploit residual connections within each CITRUS block to stabilize the training process [17]. We use the

Table 1: Traffic forecasting comparison between CITRUS and previous methods.

| | MetrLA | | | | | | | | |
|---|---|---|---|---|---|---|---|---|---|
| Method | H = 3 | | | H = 6 | | | H = 12 | | |
| | MAE | MAPE | RMSE | MAE | MAPE | RMSE | MAE | MAPE | RMSE |
| ARIMA [35] | 3.99 | 9.60% | 8.21 | 5.15 | 12.70% | 10.45 | 6.90 | 17.40% | 13.23 |
| G-VARMA [25] | 3.60 | 9.62% | 6.89 | 4.05 | 11.22% | 7.84 | 5.12 | 14.00% | 9.58 |
| GP-VAR [25] | 3.56 | 9.55% | 6.54 | 3.98 | 11.02% | 7.56 | 4.87 | 13.34% | 9.19 |
| FC-LSTM [35] | 3.44 | 9.60% | 6.30 | 3.77 | 10.90% | 7.23 | 4.37 | 13.20% | 8.69 |
| Graph Wavenet [38] | 2.69 | 6.90% | 5.15 | 3.07 | 8.37% | 6.22 | 3.53 | 10.01% | 7.37 |
| GMAN [39] | 2.94 | 7.51% | 5.89 | 3.22 | 8.92% | 6.61 | 3.68 | 10.25% | 7.49 |
| STGCN [40] | 2.88 | 7.62% | 5.74 | 3.47 | 9.57% | 7.24 | 4.59 | 12.70% | 9.40 |
| GGRNN [41] | 2.73 | 7.12% | 5.44 | 3.31 | 8.97% | 6.63 | 3.88 | 10.59% | 8.14 |
| GRUGCN [26] | 2.69 | **6.61%** | 5.15 | 3.05 | 7.96% | 6.04 | 3.62 | 9.92% | 7.33 |
| SGP [42] | 3.06 | 7.31% | 5.49 | 3.43 | 8.54% | 6.47 | 4.03 | 10.53% | 7.81 |
| GTCNN [10, 18] | **2.68** | 6.85% | 5.17 | 3.02 | 8.30% | 6.20 | 3.55 | 10.21% | 7.35 |
| **CITRUS** (Ours) | 2.70 | 6.74% | **5.14** | **2.98** | **7.78%** | **5.90** | **3.44** | **9.28%** | **6.85** |
| | PemsBay | | | | | | | | |
| Method | H = 3 | | | H = 6 | | | H = 12 | | |
| | MAE | MAPE | RMSE | MAE | MAPE | RMSE | MAE | MAPE | RMSE |
| ARIMA [35] | 1.62 | 3.50% | 3.30 | 2.33 | 5.40% | 4.76 | 3.38 | 8.30% | 6.50 |
| G-VARMA [25] | 1.88 | 4.28% | 3.96 | 2.45 | 5.42% | 4.70 | 3.01 | 7.10% | 5.83 |
| GP-VAR [25] | 1.74 | 3.45% | 3.22 | 2.16 | 5.15% | 4.41 | 2.48 | 6.18% | 5.04 |
| FC-LSTM [35] | 2.05 | 4.80% | 4.19 | 2.20 | 5.20% | 4.55 | 2.37 | 5.70% | 4.74 |
| Graph Wavenet [38] | 1.30 | 2.73% | 2.74 | 1.63 | 3.67% | 3.70 | 1.95 | 4.63% | 4.52 |
| GMAN [39] | 1.34 | 2.81% | 2.82 | 1.62 | 3.63% | 3.72 | 1.86 | 4.31% | 4.32 |
| STGCN [40] | 1.36 | 2.90% | 2.96 | 1.81 | 4.17% | 4.27 | 2.49 | 5.79% | 5.69 |
| GGRNN [41] | 1.33 | 2.83% | 2.81 | 1.68 | 3.79% | 3.94 | 2.34 | 5.21% | 5.14 |
| GRUGCN [26] | **1.21** | **2.49%** | **2.52** | 1.54 | 3.32% | 3.46 | 2.01 | 4.72% | 4.65 |
| SGP [42] | 1.33 | 2.71% | 2.84 | 1.70 | 3.61% | 3.83 | 2.24 | 5.08% | 5.19 |
| GTCNN [10, 18] | 1.25 | 2.61% | 2.66 | 1.65 | 3.82% | 3.68 | 2.27 | 5.11% | 4.99 |
| **CITRUS** (Ours) | **1.21** | 2.51% | 2.61 | **1.48** | **3.23%** | **3.28** | **1.78** | **4.08%** | **3.99** |

Table 2: Weather forecasting comparison (by rNMSE) between CITRUS and previous methods.

| Method | Molene | | | | | NOAA | | | | |
|---|---|---|---|---|---|---|---|---|---|---|
| | H = 1 | H = 2 | H = 3 | H = 4 | H = 5 | H = 1 | H = 2 | H = 3 | H = 4 | H = 5 |
| GRUGCN [26] | 0.49 | 0.56 | 0.63 | 0.70 | 0.75 | 0.15 | 0.18 | 0.24 | 0.30 | 0.37 |
| GGRNN [41] | 0.29 | 0.42 | 0.54 | 0.65 | 0.75 | 0.19 | 0.20 | 0.27 | 0.38 | 0.48 |
| SGP [42] | 0.24 | 0.37 | 0.49 | 0.59 | 0.69 | 0.39 | 0.41 | 0.43 | 0.46 | 0.49 |
| GTCNN [10, 18] | 0.39 | 0.45 | 0.52 | 0.60 | 0.68 | 0.17 | 0.19 | 0.25 | 0.31 | 0.37 |
| **CITRUS** (Ours) | **0.23** | **0.35** | **0.47** | **0.58** | **0.67** | **0.04** | **0.12** | **0.20** | **0.29** | **0.36** |

mean absolute error (MAE) as the loss function and consider a maximum of 300 epochs. The best model on the validation set is applied to the unseen test set. We compare CITRUS against previous methods using MAE, mean absolute percentage error (MAPE), and root mean squared error (RMSE).

Table 1 presents the forecasting results on the *MetrLA* and *PemsBay* datasets. We observe that, given the abundant training data in these datasets, the NN-based methods outperform the classic GSP-based methods like ARIMA [35], G-VARMA [25], and GP-VAR [25]. Similarly, the GNN-based models are superior compared to non-graph algorithms like FC-LSTM. More importantly, CITRUS outperforms state-of-the-art baselines in most metrics, especially in larger numbers of horizons ($H > 3$).

**Weather forecasting.** We also test CITRUS for weather forecasting in the *Molene* [43] and *NOAA* [44] datasets. The *Molene* dataset provides recorded temperature measurements for 744 hourly intervals over 32 measurement stations in a region in France. For the *NOAA*, which is associated with regions in the U.S., the temperature measurements were recorded for 8,579 hourly intervals over 109 stations. The pre-processing and data curation settings were similar to prior works on these datasets [10, 25]. The task here is by having the last 10 hours of measurements, we should predict the next $\{1 : 5\}$ hours. For the *Molene* dataset, the embedding dimension and the initial linear layer of

Table 3: Ablation study on comparison between the proposed CITRUS and typically ST pipelines.

| Method | *MetrLA* | | | | | | | | |
|---|---|---|---|---|---|---|---|---|---|
| | $H = 3$ | | | $H = 6$ | | | $H = 12$ | | |
| | MAE | MAPE | RMSE | MAE | MAPE | RMSE | MAE | MAPE | RMSE |
| TTS | 2.73 | 6.78% | 5.21 | 3.10 | 8.02% | 6.15 | 3.67 | 10.05% | 7.46 |
| STT | 2.72 | 6.74% | 5.19 | 3.07 | 7.94% | 6.08 | 3.65 | 10.97% | 7.37 |
| CTTS | **2.70** | 6.69% | 5.20 | 3.05 | 7.93% | 6.18 | 3.61 | 9.75% | 7.51 |
| CSTT | **2.70** | **6.66%** | 5.22 | 3.06 | 7.92% | 6.19 | 3.63 | 9.92% | 7.48 |
| **CITRUS** (Ours) | **2.70** | 6.74% | **5.14** | **2.98** | **7.78%** | **5.90** | **3.44** | **9.28%** | **6.85** |

Table 4: Training time (per epoch) and forecasting results vs. number of selected eig-eiv on *MetrLA*.

| | $k = 2$ | $k = 5$ | $k = 10$ | $k = 50$ | $k = 100$ | $k = 150$ | $k = 200$ |
|---|---|---|---|---|---|---|---|
| MAE | 2.80 | 2.74 | 2.72 | 2.71 | 2.71 | 2.70 | 2.70 |
| Training time (seconds) | 2.52 | 2.91 | 2.96 | 3.86 | 5.28 | 6.36 | 7.78 |

the CITRUS blocks are 16, while no MLP modules have been used. Regarding *NOAA*, we set the dimension and the initial linear layer of the CITRUS blocks as 16, and use 3-layer MLPs for channel mixing in each block. The CITRUS blocks have the last activation as a Leaky ReLU. We use MSE as the loss function and consider root-normalized MSE (rNMSE) as the evaluation metric.

Table 2 illustrates the results of the weather forecasting task. We observe that CITRUS show superior forecasting performance compared to previous methods in all forecasting horizons. We note that the SGP model [42], which is a scalable architecture, is the second-best performing method in *Molene*, possibly due to the small scale of this dataset. On the contrary, the *NOAA* dataset is larger, so models with higher parameter budgets perform better.

### 4.4 Ablation Study and Hyperparameter Sensitivity Analysis

**Ablation study.** The ablation study concerns the CITRUS modules responsible for *jointly* learning spatiotemporal couplings. To this end, we use two well-known architectures: TTS and STT. Therefore, we evaluate four possible configurations: i) TTS, where we first apply an RNN-based network (here, GRU), and then the outputs are fed into a regular GNN; ii) STT, which is exactly the opposite of the TTS; iii) Continuous TTS (CTTS), where we replace the GNN in TTS with a CGNN; and iv) Continous STT (CSTT), which is the opposite of CTTS. We report the results of this ablation study on the *MetrLA* dataset in Table 3. We observe that CITRUS has superior results probably due to learning *joint* (and not sequential) spatiotemporal couplings by modeling these dependencies using product graphs with learnable receptive fields. We also observe that CTTS and CSTT outperform their discrete counterparts TTS and STT, probably due to the learning of adaptive graph neighborhoods instead of relying on 1-hop connections in regular GNNs.

**Hyperparameter sensitivity analysis.** The sensitivity analysis is related to the number of selected eigenvector-eigenvalue (eig-eiv) pairs of the factor Laplacians. We select eigenvector-eigenvalue pairs in $k \in \{2, 10, 50, 100, 150, 200\}$ out of 207 components in the *MetrLA* dataset. Table 4 presents the forecasting results in MAE along with the training time (per epoch) of this ablation study. We observe that selecting 50 eigenvector-eigenvalue pairs is enough to get good forecasting performance while having a fast training time. This experiment illustrates the advantages of CITRUS for accelerating training and keeping the performance as consistent as possible.

## 5 Conclusion and Limitations

In this paper, we proposed CITRUS, a novel model for jointly learning multidomain couplings from product graph signals based on tensorial PDEs on graphs (TPDEGs). We modeled these representations as separable continuous heat graph kernels as solutions to the TPDEG. Therefore, we showed that the underlying graph is actually the Cartesian product of the domain-specific factor

graphs. We rigorously studied the stability and over-smoothing aspects of CITRUS theoretically and experimentally. Finally, as a proof of concept, we use CITRUS to tackle the traffic and weather spatiotemporal forecasting tasks on public datasets, illustrating the superior performance of our model compared to state-of-the-art methods.

An interesting future direction for our framework could involve adapting it to handle other types of graph products, such as Kronecker and Strong graph products [23, 24]. Future efforts will also focus on finding tighter and more general upper bounds in the stability and over-smoothing analyses. For example, we will explore the possible relationships between the size of the factor graphs and these properties, either in a general form or for specific well-studied structures, such as ER graphs. Additionally, we plan to investigate more challenging real-world applications of the proposed framework, especially in scenarios involving more than two factor graphs.

## Acknowledgments

This work has been carried out at the Energy4Climate Interdisciplinary Center (E4C) of IP Paris and Ecole Nationale des Ponts et Chaussées, which is in part supported by 3rd Programme d'Investissements d'Avenir (ANR-18-EUR-0006-02), and by the Foundation of Ecole Polytechnique with the private sponsor Fonds Ifker pour le Climat, financed by Stéphane y Agnès Ifker. This work was also supported by the center Hi! PARIS and ANR (French National Research Agency) under the JCJC project GraphIA (ANR-20-CE23-0009-01).

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

# Appendix

The appendix contains the proof of the claimed theoretical statements (Section A), descriptions of the compared methods (Section C), the standard deviation of the results (Section D), homophily-heterophily trade-off (Section E), the case of inexact Cartesian product graphs (Section F), how to choose the appropriate number of eigenvalue-eigenvector pairs (Section G), the effect of graph receptive fields on the over-smoothing (Section H) and hyperparameter details (Section I), respectively.

## A   Proofs

### A.1   Proof of Theorem 3.2

*Proof.* For obtaining the derivative of $\frac{\partial \underline{\mathbf{U}}_t}{\partial t}$ from $e^{-t\mathbf{L}_p}$ (for $p = 1, \ldots, P$) in (4), we first express $\frac{\partial \underline{\mathbf{U}}_t}{\partial t}$ from the claimed solution in (4) as follows:

$$\frac{\partial \underline{\mathbf{U}}_t}{\partial t} = \sum_{p=1}^{P} \underline{\mathbf{U}}_{t,p}, \tag{19}$$

where

$$\underline{\mathbf{U}}_{t,p} := \underline{\mathbf{U}}_0 \times_1 e^{-t\mathbf{L}_1} \times_2 \ldots \times_{p-1} e^{-t\mathbf{L}_{p-1}} \times_p \left(-\mathbf{L}_p e^{-t\mathbf{L}_p}\right) \times_{p+1} e^{-t\mathbf{L}_{p+1}} \times_{p+2} \ldots \times_P e^{-t\mathbf{L}_P}. \tag{20}$$

Now, by performing mode-$p$ unfolding operation on $\underline{\mathbf{U}}_{t,p}$, we have:

$$\underline{\mathbf{U}}_{t,p(p)} = -\mathbf{L}_p e^{-t\mathbf{L}_p} \underline{\mathbf{U}}_{0(p)} \left(e^{-t\mathbf{L}_P} \otimes e^{-t\mathbf{L}_{P-1}} \otimes \ldots e^{-t\mathbf{L}_{p+1}} \otimes e^{-t\mathbf{L}_{p-1}} \otimes \ldots \otimes e^{-t\mathbf{L}_1}\right)^{\top}. \tag{21}$$

On the other hand:

$$\underline{\mathbf{U}}_{t(p)} = e^{-t\mathbf{L}_p} \underline{\mathbf{U}}_{0(p)} \left(e^{-t\mathbf{L}_P} \otimes \ldots e^{-t\mathbf{L}_{p+1}} \otimes e^{-t\mathbf{L}_{p-1}} \otimes \ldots \otimes e^{-t\mathbf{L}_1}\right)^{\top}. \tag{22}$$

Therefore, by combining (21) and (22), one can write:

$$\underline{\mathbf{U}}_{t,p(p)} = -\mathbf{L}_p \underline{\mathbf{U}}_{t(p)}, \tag{23}$$

which is the mode-$p$ unfolding of the following equation:

$$\underline{\mathbf{U}}_{t,p} = -\underline{\mathbf{U}}_t \times_p \mathbf{L}_p, \tag{24}$$

and the proof is completed. $\qquad \square$

### A.2   Proof of Theorem 3.3

*Proof.* First, we prove the following lemma about the separability the heat kernels define on product graphs w.r.t. the factor graphs:

**Lemma A.1.** *The exponential graph kernel $e^{t\mathbf{L}_\diamond}$ on a Cartesian product graph $\mathbf{L}_\diamond := \oplus_{p=1}^{P} \mathbf{L}_p$ can be Kronecker-factored into its factor-based graph kernels as $e^{t\mathbf{L}_\diamond} = \otimes_{p=1}^{P} e^{t\mathbf{L}_p}$.*

*Proof.* Using the pairwise summation property of the factor eigenvalues in a Cartesian product, one can write:

$$\begin{aligned}
e^{t\mathbf{L}_\diamond} &= e^{t\left(\oplus_{p=1}^{P} \mathbf{L}_p\right)} \\
&= \left(\otimes_{p=1}^{P} \mathbf{V}_p\right) e^{t\left(\oplus_{p=1}^{P} \mathbf{\Lambda}_p\right)} \left(\otimes_{p=1}^{P} \mathbf{V}_p\right)^{\top} \\
&= \left(\otimes_{p=1}^{P} \mathbf{V}_p\right) \left(\otimes_{p=1}^{P} e^{t\mathbf{\Lambda}_p}\right) \left(\otimes_{p=1}^{P} \mathbf{V}_p\right)^{\top} \\
&= \otimes_{p=1}^{P} e^{t\mathbf{L}_p},
\end{aligned} \tag{25}$$

where $\mathbf{V}_p$, $\lambda_i^{(p)}$, and $\mathbf{\Lambda}_p$ are the eigenmatrix, $i$th eigenvalue, and the diagonal eigenvalue matrix corresponding to the $p$th Laplacian $\mathbf{L}_p$. Note that, in the related literature, this property has also been proved from different points of view, *e.g.*, [34]. $\qquad \square$

Next, by performing mode-$(P+1)$ unfolding on both sides of (5), one can write:

$$(\mathcal{L}_l(\underline{\mathbf{U}}_l))_{(P+1)} = \mathbf{W}_l^\top \underline{\mathbf{U}}_{l(P+1)} \left[\downarrow \otimes_{p=1}^P e^{-t_l \mathbf{L}_p}\right]^\top$$

$$\rightarrow \left[(\mathcal{L}_l(\underline{\mathbf{U}}_l))_{(P+1)}\right]^\top = \overbrace{\left[\downarrow \otimes_{p=1}^P e^{-t_l \mathbf{L}_p}\right]}^{e^{-t_l \bar{\mathbf{L}}_\diamond}} \underline{\mathbf{U}}_{l(P+1)}^\top \mathbf{W}_l. \tag{26}$$

$\square$

## A.3 Proof of Proposition 3.6

*Proof.* The proof is presented by construction. Therefore, we first show it is true for the case of $P = 2$ as:

$$\begin{aligned}
\tilde{\mathbf{A}} &= \tilde{\mathbf{A}}_1 \otimes \mathbf{I}_2 + \mathbf{I}_1 \otimes \tilde{\mathbf{A}}_2 \\
&= (\mathbf{A}_1 + \mathbf{E}_1) \otimes \mathbf{I}_2 + \mathbf{I}_1 \otimes (\mathbf{A}_2 + \mathbf{E}_2) \\
&= \overbrace{(\mathbf{A}_1 \otimes \mathbf{I}_2 + \mathbf{I}_1 \otimes \mathbf{A}_2)}^{\mathbf{A}} + \overbrace{(\mathbf{E}_1 \otimes \mathbf{I}_2 + \mathbf{I}_1 \otimes \mathbf{E}_2)}^{\mathbf{E}} \\
&= \mathbf{A} + \mathbf{E}.
\end{aligned} \tag{27}$$

Then, by assuming the theorem holds for the case of $P = K$ and the definitions of $\tilde{\mathbf{A}}'_K := \oplus_{p=1}^K \tilde{\mathbf{A}}_p$ and $\tilde{\mathbf{A}}'_K := \oplus_{p=1}^K \tilde{\mathbf{A}}_p$, we next show it also holds for $P = K+1$ as follows:

$$\begin{aligned}
\tilde{\mathbf{A}} &= \oplus_{p=1}^{K+1} \tilde{\mathbf{A}}_p = \overbrace{(\oplus_{p=1}^K \tilde{\mathbf{A}}_p)}^{\tilde{\mathbf{A}}'_K} \oplus \tilde{\mathbf{A}}_{K+1} = \tilde{\mathbf{A}}'_K \otimes \mathbf{I}_2 + \mathbf{I}_1 \otimes \tilde{\mathbf{A}}_{K+1} \\
&= (\mathbf{A}'_K + \mathbf{E}'_K) \otimes \mathbf{I}_2 + \mathbf{I}_1 \otimes (\mathbf{A}_{K+1} + \mathbf{E}_{K+1}) \\
&= \overbrace{(\mathbf{A}'_K \otimes \mathbf{I}_2 + \mathbf{I}_1 \otimes \mathbf{A}_{K+1})}^{\mathbf{A}=\oplus_{p=1}^{K+1} \mathbf{A}_p} + \overbrace{(\mathbf{E}'_K \otimes \mathbf{I}_2 + \mathbf{I}_1 \otimes \mathbf{E}_{K+1})}^{\mathbf{E}=\oplus_{p=1}^{K+1} \mathbf{E}_p} \\
&= \mathbf{A} + \mathbf{E}.
\end{aligned} \tag{28}$$

For proving the upper bound of the norm of $\mathbf{E}$, in a similar approach to the previous proof, we first prove for the case of $P = 2$ as:

$$\begin{aligned}
|||\mathbf{E}||| &= |||\mathbf{E}_1 \otimes \mathbf{I}_2 + \mathbf{I}_1 \otimes \mathbf{E}_2||| \leq |||\mathbf{E}_1 \otimes \mathbf{I}_2||| + |||\mathbf{I}_1 \otimes \mathbf{E}_2||| \\
&\leq \overbrace{|||\mathbf{E}_1|||}^{\varepsilon_1} \cdot \overbrace{|||\mathbf{I}_2|||}^{1} + \overbrace{|||\mathbf{I}_1|||}^{1} \cdot \overbrace{|||\mathbf{E}_2|||}^{\varepsilon_2} = \varepsilon_1 + \varepsilon_2.
\end{aligned} \tag{29}$$

Using a similar approach to the proof in (28), we assume the theorem holds for $P = K$ as $|||\mathbf{E}||| \leq \sum_{p=1}^K \varepsilon_p$, and, based on this information, then, we prove the theorem for the case of $P = K+1$ as follows:

$$\begin{aligned}
|||\mathbf{E}||| &= |||\mathbf{E}'_K \otimes \mathbf{I}_2 + \mathbf{I}_1 \otimes \mathbf{E}_{K+1}||| \leq |||\mathbf{E}'_K \otimes \mathbf{I}_2||| + |||\mathbf{I}_1 \otimes \mathbf{E}_{K+1}||| \\
&\leq \overbrace{|||\mathbf{E}'_K|||}^{\sum_{p=1}^K \varepsilon_p} \cdot \overbrace{|||\mathbf{I}_2|||}^{1} + \overbrace{|||\mathbf{I}_1|||}^{1} \cdot \overbrace{|||\mathbf{E}_{K+1}|||}^{\varepsilon_{K+1}} = \sum_{p=1}^{K+1} \varepsilon_p,
\end{aligned} \tag{30}$$

which concludes the proof. $\square$

## A.4 Proof of Theorem 3.7

*Proof.* The poof can be straightforwardly obtained by applying Proposition 1 in [30] based on the obtained results in Proposition 3.6. $\square$

## A.5 Proof of Lemma 3.9

*Proof.* By construction, we first prove the theorem for the case of $P = 2$. The defined Laplacian $\hat{\mathbf{L}}$ in (15) can be rewritten as:

$$
\begin{aligned}
\hat{\mathbf{L}} = \mathbf{I} - \hat{\mathbf{A}} &= \mathbf{I} - \left( \frac{\hat{\mathbf{A}}_1}{2} \otimes \mathbf{I}_2 + \mathbf{I}_1 \otimes \frac{\hat{\mathbf{A}}_2}{2} \right) \\
&= \frac{1}{2} \left( 2\mathbf{I} - \left( \hat{\mathbf{A}}_1 \otimes \mathbf{I}_2 + \mathbf{I}_1 \otimes \hat{\mathbf{A}}_2 \right) \right) \\
&= \frac{1}{2} \left( [\mathbf{I} - (\hat{\mathbf{A}}_1 \otimes \mathbf{I}_2)] + [\mathbf{I} - (\mathbf{I}_1 \otimes \hat{\mathbf{A}}_2)] \right) \\
&= \frac{1}{2} \left( \left[ \mathbf{I}_1 \otimes \mathbf{I}_2 - (\hat{\mathbf{A}}_1 \otimes \mathbf{I}_2) \right] + \left[ \mathbf{I}_1 \otimes \mathbf{I}_2 - (\mathbf{I}_1 \otimes \hat{\mathbf{A}}_2) \right] \right) \\
&= \frac{1}{2} \left( \left[ (\mathbf{I}_1 - \hat{\mathbf{A}}_1) \otimes \mathbf{I}_2 \right] + \left[ \mathbf{I}_1 \otimes (\mathbf{I}_2 - \hat{\mathbf{A}}_2) \right] \right) \\
&= \frac{1}{2} \left( \hat{\mathbf{L}}_1 \otimes \mathbf{I}_2 + \mathbf{I}_1 \otimes \hat{\mathbf{L}}_2 \right) \\
&= \frac{1}{2} \left( \hat{\mathbf{L}}_1 \oplus \hat{\mathbf{L}}_2 \right) \\
&= \left( \frac{\hat{\mathbf{L}}_1}{2} \right) \oplus \left( \frac{\hat{\mathbf{L}}_2}{2} \right),
\end{aligned}
\tag{31}
$$

where we used the following property of Kronecker products: $\alpha(A \otimes B) = (\alpha A) \otimes B = A \otimes (\alpha B)$ [45]. Next, by assuming that the theorem holds for $P = K$ and the definition of $\hat{\mathbf{A}}'_K := \frac{1}{K}$ and $\oplus_{p=1}^{K} \hat{\mathbf{A}}_p := \oplus_{p=1}^{K} \left( \frac{\hat{\mathbf{A}}_p}{K} \right)$, we prove it for $P = K + 1$:

$$
\begin{aligned}
\hat{\mathbf{L}} = \mathbf{I} - \hat{\mathbf{A}} &= \mathbf{I} - \frac{1}{K+1} \left( K\hat{\mathbf{A}}'_K \oplus \hat{\mathbf{A}}_{K+1} \right) \\
&= \mathbf{I} - \left( \frac{K\hat{\mathbf{A}}'_K}{K+1} \otimes \mathbf{I}_2 + \mathbf{I}_1 \otimes \frac{\hat{\mathbf{A}}_{K+1}}{K+1} \right) \\
&= \frac{1}{K+1} \left( (K+1)\mathbf{I} - \left( K\hat{\mathbf{A}}'_K \otimes \mathbf{I}_2 + \mathbf{I}_1 \otimes \hat{\mathbf{A}}_{K+1} \right) \right) \\
&= \frac{1}{K+1} \left( K[\mathbf{I} - (\hat{\mathbf{A}}'_K \otimes \mathbf{I}_2)] + [\mathbf{I} - (\mathbf{I}_1 \otimes \hat{\mathbf{A}}'_{K+1})] \right) \\
&= \frac{1}{K+1} \left( \left[ \mathbf{I}_1 \otimes \mathbf{I}_2 - (\hat{\mathbf{A}}'_K \otimes \mathbf{I}_2) \right] + \left[ \mathbf{I}_1 \otimes \mathbf{I}_2 - (\mathbf{I}_1 \otimes \hat{\mathbf{A}}_2) \right] \right) \\
&= \frac{1}{K+1} \left( K \left[ (\mathbf{I}_1 - \hat{\mathbf{A}}'_K) \otimes \mathbf{I}_2 \right] + \left[ \mathbf{I}_1 \otimes (\mathbf{I}_2 - \hat{\mathbf{A}}_2) \right] \right) \\
&= \frac{1}{K+1} \left( K \left[ \oplus_{p=1}^{K} \left( \frac{\hat{\mathbf{L}}_p}{K} \right) \right] \otimes \mathbf{I}_2 + \mathbf{I}_1 \otimes \hat{\mathbf{L}}_2 \right) \\
&= \frac{1}{K+1} \left( \oplus_{p=1}^{K+1} \left( \hat{\mathbf{L}}_p \right) \right) \\
&= \oplus_{p=1}^{K+1} \left( \frac{\hat{\mathbf{L}}_p}{K+1} \right).
\end{aligned}
\tag{32}
$$

Besides, by the addition rule for the eigenvalues of a product graph adjacency or Laplacian ($\lambda_{ij}^{\diamond} = \lambda_i^{(1)} + \lambda_j^{(2)}$), it can be easily seen that the eigenvalues of the normalized product adjacency ($\hat{\mathbf{W}}$) and Laplacian ($\hat{\mathbf{L}}$) in (31) are in the intervals of $[-1, 1]$ and $[0, 2]$, respectively, quite similar to the case for the normalized factor graphs, which concludes the proof. $\square$

### A.6 Proof of Theorem 3.10

*Proof.* First, Note that Using $\tilde{\mathbf{x}}$ as the Graph Fourier Transform (GFT) [13] of $\mathbf{x}$ w.r.t. $\hat{\mathbf{L}}$ (with eigenvalues $\{\lambda_i\}_{i=1}^N$), one can write [15]:

$$E(\mathbf{x}) = \mathbf{x}^\top \hat{\mathbf{L}} \mathbf{x} = \sum_{i=1}^N \lambda_i \tilde{x}_i^2. \tag{33}$$

Next, by defining $\lambda$ as the smallest non-zero eigenvalue of the Laplacian $\hat{\mathbf{L}}$, the following lemma characterizes the over-smoothing aspects of applying a heat kernel in the simplest case.

**Lemma A.2.** *We have:*

$$E(e^{-\hat{\mathbf{L}}}\mathbf{x}) \le e^{-2\lambda} E(\mathbf{x}). \tag{34}$$

*Proof.* By considering the EVD forms of $\hat{\mathbf{L}} = \mathbf{V}\boldsymbol{\Lambda}\mathbf{V}^\top$ and $e^{-\hat{\mathbf{L}}} = \mathbf{V}e^{-\boldsymbol{\Lambda}}\mathbf{V}^\top$, one can write

$$E(e^{-\hat{\mathbf{L}}}\mathbf{x})$$

$$= \mathbf{x}^\top \overbrace{e^{-\hat{\mathbf{L}}^\top}}^{\mathbf{V}e^{-\boldsymbol{\Lambda}}\mathbf{V}^\top} \overbrace{\hat{\mathbf{L}}}^{\mathbf{V}\boldsymbol{\Lambda}\mathbf{V}^\top} \overbrace{e^{-\hat{\mathbf{L}}}}^{\mathbf{V}e^{-\boldsymbol{\Lambda}}\mathbf{V}^\top} \mathbf{x} = \sum_{i=1}^N \lambda_i \tilde{x}_i^2 e^{-2\lambda_i} \le e^{-2\lambda} \left( \sum_{i=1}^N \lambda_i \tilde{x}_i^2 \right) = e^{-2\lambda} E(\mathbf{x}). \tag{35}$$

Note that in the above proof, we ruled out the zero eigenvalues since they are useless in analyzing Dirichlet energy. $\square$

Then by considering the following lemmas from [15]:

**Lemma A.3.** *(Lemma 3.2 in [15]).* $E(\mathbf{X}\mathbf{W}) \le \|\mathbf{W}^\top\|_2^2 E(\mathbf{X})$.

**Lemma A.4.** *(Lemma 3.3 in [15]). For ReLU and Leaky-ReLU nonlinearities* $E(\sigma(\mathbf{X})) \le E(\mathbf{X})$.

Therefore, by combining the previous concepts and considering that the minimum non-zero eigenvalue of $\hat{\mathbf{L}}^{(t)}$ in (16) is $\frac{1}{P}t^{(m)}\lambda^{(m)}$ with $m = \arg\min_i t^{(i)}\lambda^{(i)}$, it can be said that:

**Theorem A.5.** *For any* $l \in \mathbb{N}_+$*, we have* $E(f_l(\mathbf{X})) \le s_l e^{-\frac{2}{P}\tilde{t}\tilde{\lambda}} E(\mathbf{X})$*, where* $s_l := \prod_{h=1}^{H_l} s_{lh}$ *and* $s_{lh}$ *is the square of the maximum singular value of* $\mathbf{W}_{lh}^\top$*. Besides,* $\tilde{\lambda} = \lambda^{(m)}$ *and* $\tilde{t} = t^{(m)}$ *with* $m = \arg\min_i t^{(i)}\lambda^{(i)}$*, where* $\lambda^{(i)}$ *is the smallest non-zero eigenvalue of* $\hat{\mathbf{L}}_i$*.*

Afterward, we can state our final corollary as follows:

**Corollary A.6.** *Let* $s := \sup_{l \in \mathbb{N}_+} s_l$*. We have:*

$$E(\mathbf{X}^{(l)}) \le s^l e^{-\frac{2}{P}l\tilde{t}\tilde{\lambda}} E(\mathbf{X}) = e^{l\left(\ln s - \frac{2}{P}\tilde{t}\tilde{\lambda}\right)} E(\mathbf{X}). \tag{36}$$

*So,* $E(\mathbf{X}^{(l)})$ *exponentially converges to 0, when* $\lim_{l \to \infty} e^{l\left(\ln s - \frac{2}{P}\tilde{t}\tilde{\lambda}\right)} = 0$*, i.e.,* $\ln s - \frac{2}{P}\tilde{t}\tilde{\lambda} < 0$*.*

$\square$

## B Experiments on more than Two Factor Graphs

We consider the node regression task similar to the settings described in Section 4.1 but for three ER graphs with $p_{\text{ER}}^{(1)} = p_{\text{ER}}^{(1)} = 0.3$, and varying $p_{\text{ER}}^{(3)} \in \{0.1, 0.3\}$ to monitor the performance in two different connectivity scenarios. Note that here we do not perturb factor graphs but, to make the experiment more challenging, the SNR on the graph data is set to SNR=0. Table 5 provides the results across a varying number of layers, which shows that the proposed framework's performance is resistant to adding layers, especially in the case of more sparse graphs, *i.e.*, $p_{\text{ER}}^{(3)} = 0.1$. On the other hand, the best results were obtained in $l = 1, 2, 4$, which are the nearest numbers to the actual number of layers in the true generative process which was 3. These observations are validated by comparing the results with the GCN, in which the over-smoothing and performance degradation happen severely faster than CITRUS.

Table 5: Experiments on more than two factor graphs ($p_{\mathrm{ER}}^{(1)} = p_{\mathrm{ER}}^{(2)} = 0.3$).

| | $l = 1$ | $l = 2$ | $l = 4$ | $l = 8$ | $l = 16$ | $l = 32$ |
|---|---|---|---|---|---|---|
| GCN, $p_{\mathrm{ER}}^{(3)} = 0.1$ | 0.185±0.132 | 0.241±0.096 | 0.371±0.132 | 0.458±0.132 | 0.494±0.147 | 0.552±0.118 |
| CITRUS, $p_{\mathrm{ER}}^{(3)} = 0.1$ | 0.141±0.075 | 0.137±0.082 | 0.150±0.091 | 0.198±0.106 | 0.215±0.124 | 0.276±0.120 |
| GCN, $p_{\mathrm{ER}}^{(3)} = 0.3$ | 0.253±0.211 | 0.296±0.113 | 0.365±0.123 | 0.429±0.097 | 0.517±0.143 | 0.67±0.153 |
| CITRUS, $p_{\mathrm{ER}}^{(3)} = 0.3$ | 0.188±0.094 | 0.182±0.095 | 0.193±0.096 | 0.230±0.089 | 0.242±0.096 | 0.336±0.107 |

Table 6: Standard deviation of traffic forecasting comparison in Table 1.

| | | | | | | | | | |
|---|---|---|---|---|---|---|---|---|---|
| | | | | *MetrLA* | | | | | |
| Method | $H = 3$ | | | $H = 6$ | | | $H = 12$ | | |
| | MAE | MAPE | RMSE | MAE | MAPE | RMSE | MAE | MAPE | RMSE |
| **CITRUS** (Ours) | 0.015 | 0.009 | 0.008 | 0.007 | 0.012 | 0.003 | 0.002 | 0.005 | 0.004 |
| | | | | *PemsBay* | | | | | |
| Method | $H = 3$ | | | $H = 6$ | | | $H = 12$ | | |
| | MAE | MAPE | RMSE | MAE | MAPE | RMSE | MAE | MAPE | RMSE |
| **CITRUS** (Ours) | 0.009 | 0.008 | 0.003 | 0.013 | 0.006 | 0.005 | 0.008 | 0.011 | 0.008 |

# C  Descriptions of the Compared Methods

- ARIMA [35]: uses Kalman filter to implement auto-regressive integrated moving average framework, but processes data in a timestep-wise manner.

- G-VARMA [25]: generalizes the vector auto-regressive moving average (VARMA) to the case of considering graph-based data by imposing some statistical assumptions on the model and relying on the spatial graph.

- GP-VAR [25]: adapts AR model by spatial graph polynomial filters, with fewer number of learnable parameters.

- FC-LSTM [35]: a hybrid of fully connected MLPs and LSTM cells that considers each time step separately by assigning one LSTM cell to them.

- Graph WaveNet [38]: a hybrid adapted convolutional and attentional network with GNNs for ST forecasting.

- GMAN [39]: exploits multiple ST attention layers within an encoder-decoder framework.

- STGCN [40]: a hybrid of typical GCNs and 1D convolutional blocks for ST forecasting.

- GGRNN [41]: generalizes the typical RNNs by adapting GCN blocks instead of linear MLPs.

- GRUGCN [26]: a TTS framework, which first processes the temporal information by a GRU cell and then learns spatial representations by applying typical GCN blocks.

- SGP [42]: a scalable graph-based ST framework for considerably reducing the number of learnable parameters. Note that we used the version without positional embedding blocks.

# D  Standard Deviation of the Results

The standard deviation of traffic forecasting in Table 1 and weather forecasting regarding the proposed framework in Table 2 are provided in Tables 6 and 7.

Table 7: Standard deviation of weather forecasting comparison in Table 2.

| Method | Molene | | | | | NOAA | | | | |
|---|---|---|---|---|---|---|---|---|---|---|
| | $H=1$ | $H=2$ | $H=3$ | $H=4$ | $H=5$ | $H=1$ | $H=2$ | $H=3$ | $H=4$ | $H=5$ |
| **CITRUS** (ours) | 0.0174 | 0.0155 | 0.0132 | 0.0107 | 0.0086 | 0.0034 | 0.0030 | 0.0026 | 0.0024 | 0.0023 |

Table 8: Intra and Inter-homophily measures on the real-world graphs of *MetrLA* and *PemsBay* datasets.

| Dataset | Intra-graph homophily $\boldsymbol{\pi}_{v_i}^s$ | | | Inter-graph homophily $\boldsymbol{\pi}_{v_i}^T$ | | |
|---|---|---|---|---|---|---|
| | $p_s$ | $q_p$ | $q_n$ | $p_s$ | $q_p$ | $q_n$ |
| *MetrLA* | 0.2273 | 0.4732 | 0.2995 | 0.3325 | 0.4920 | 0.1755 |
| *PemsBay* | 0.1073 | 0.5912 | 0.3015 | 0.2399 | 0.6863 | 0.0738 |
| *Molene* | 0.0148 | 0.6248 | 0.3602 | 0.0152 | 0.5545 | 0.4301 |
| *NOAA* | 0.1249 | 0.5345 | 0.3405 | 0.2862 | 0.4351 | 0.2786 |

## E   Homophily-Heterophily Trade-off across the Studied Real Datasets

Regarding the homophily-heterophily trade-off, first, we should mention that measuring the homophily-heterophily indices is way more challenging in spatiotemporal settings rather than in single-graph scenarios [46], since we are facing two intra-graph (*i.e.*, the spatial dimension) and inter-graph (*i.e.*, the temporal domain of changing time series characteristics) aspects. In this way, in the recent pioneer literature, they considered *MetrLA*, *PemsBay*, and temperature datasets (like *Molene* and *NOAA* used in our paper) as complicated in terms of homophily-heterophily behaviors. We provided the measured homophily-heterophily metrics (full details in [46]) in Table 8. As observed in this table, due to the wide range of variability across the used datasets, we have actually evaluated our performance on various datasets in terms of homophily-heterophily levels. For instance, *Molene* acts severely heterophilic in both intra and inter-graph domains, while *NOAA* acts more homophilic than *Molene* in the inter-graph domain.

## F   The Case of Inexact Cartesian Product Modeling

This scenario lies under the general umbrella of inaccurate adjacencies [30], which has been rigorously studied in the stability analysis in Section 3.2. However, in a simplified and insightful case of facing graph products other than Cartesian, we consider two well-known graph products, *i.e.*, Strong and Kronecker products [23, 24]. Consider the true graph is actually a Strong product graph $\mathbf{A} = \mathbf{A}_1 \otimes \mathbf{A}_2 + \mathbf{A}_1 \otimes \mathbf{I}_2 + \mathbf{I}_1 \otimes \mathbf{A}_2$, and one mistakenly considers it as Cartesian product $\tilde{\mathbf{A}} = \mathbf{A}_1 \otimes \mathbf{I}_2 + \mathbf{I}_1 \otimes \mathbf{A}_2$. Then, $\mathbf{E} = \mathbf{A} - \tilde{\mathbf{A}} = \mathbf{A}_1 \otimes \mathbf{A}_2$, where $\|\mathbf{E}\| = \lambda_{max}^{(1)} \lambda_{max}^{(2)}$. So, the approximation error depends on the multiplication of the factor graph eigenspaces. Similarly, in the case of Kronecker product graphs $\tilde{\mathbf{A}} = \mathbf{A}_1 \otimes \mathbf{A}_2$, it obtains $\|\mathbf{E}\| \leq \lambda_{max}^{(1)} + \lambda_{max}^{(2)} + \lambda_{max}^{(1)} \lambda_{max}^{(2)}$. So, in this case, the summation of the factor eigenspaces matters too. Briefly, the approximation error bound depends on the factor graph spectrums.

## G   Choosing Appropriate $k$

We can choose an appropriate $k$ using two methodologies: supervised and unsupervised. For the supervised option, we can use cross-validation, as we did in our results. For the unsupervised method, we can analyze the Laplacian matrices for a low-rank approximation task. For example, in Figure 4, we plot the explained variances of the principal components of the spatial Laplacian (*i.e.*, $\|\mathbf{v}\lambda\mathbf{v}^\top\|_F^2$ for $\mathbf{v}$ and $\lambda$ being the eigenvectors and eigenvalues) on the *MetrLA* dataset. We observe a strong concentration of variance in a few eigenvalues. We also observe that we can capture almost 80% of the explained variance by only relying on 50 components.

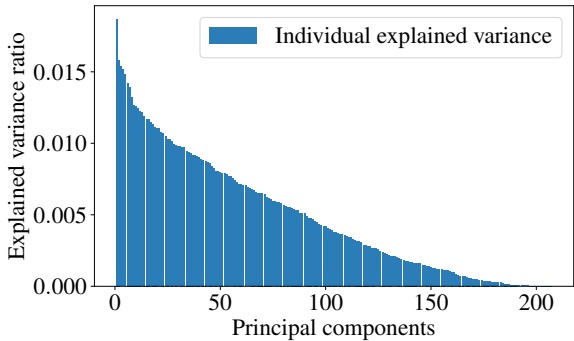

Figure 4: Explained variance ratio vs. selected principal components.

Table 9: Details of the training settings and hyperparameters, *i.e.*, $T$ (auto-regressive order), $emb$ (dimension of embedding size in the spatiotemporal encoder), $hid$ (dimension of linear mapping size in the spatiotemporal encoder), $F_{\text{MLP}}$ (dimension of linear mapping size in the MLP layers), $n_{\text{CITRUS}}$ (number of CITRUS blocks), $F$ (second dimension of $\mathbf{W}_l$ in (6)), $k_i$ (number of selected eigenvalue-eigenvector pairs in the $i$-th factor graph), $lr$ (learning rate), $n_{\text{batch}}$ (batch size), and $n_{\text{epochs}}$ (number of epochs).

| Dataset | $T$ | $emb$ | $hid$ | $F_{\text{MLP}}$ | $n_{\text{CITRUS}}$ | $F$ | $k_1$ | $k_2$ | $lr$ | $n_{\text{batch}}$ | $n_{\text{epochs}}$ |
|---------|-----|-------|-------|------------------|---------------------|-----|-------|-------|------|--------------------|---------------------|
| *MetrLA* | 6 | 16 | 32 | 64 | 3 | 64 | 205 | 4 | 0.01 | 2048 | 300 |
| *PemsBay* | 6 | 16 | 32 | 64 | 3 | 64 | 323 | 4 | 0.01 | 2048 | 300 |
| *Molene* | 10 | 16 | 16 | 16 | 3 | 16 | 30 | 8 | 0.001 | 64 | 1500 |
| *NOAA* | 10 | 4 | 4 | 4 | 3 | 4 | 107 | 8 | 0.01 | 256 | 400 |

## H   The Effect of the Graph Receptive Fields on Controlling Over-smoothing

From a theoretical point of view, based on eq. (18), one intuitive way to alleviate or at least slow down the over-smoothing phenomena is to keep $\ln s - \frac{2}{P}\tilde{t}\tilde{\lambda}$ positive, which tends to $\tilde{t} < \frac{P}{2\tilde{\lambda}}\ln s$. This clarifies why one should not increase the node-neighborhood especially in the case of facing a connected graph or increasing the number of layers. This finding is also compatible with the pioneering research about the over-smoothing concept, *e.g.*, [14]. From an experimental point of view, and if we consider the graph receptive field $t$ as a hyperparameter, we have designed an experiment to validate the claimed statement. Here, we vary $t \in \{0.1, 1, 5, 10, 20\}$ and monitor the over-smoothing process in Figure 5. In this figure, $b = \frac{P}{2\tilde{\lambda}}\ln s$, and as it is observed, for values of $t$ higher than $b$ the over-smoothing phenomena with increasing the number of layers is happening way faster than the other ones, which validates the theoretical discussion, too. Apart from this, the weight normalization technique [14,15] has also been mentioned as a helpful action but it hinders the theoretical foundations of CITRUS. Note that, in practice, only escaping from the over-smoothing is not enough to have good performance. Therefore, for $t < b$, one might need to perform other kinds of analyses, *e.g.*, over-squashing analysis.

## I   Hyperparameter Details

The detailed hyperparameters (optimized by cross-validation on the validation data) and/or training settings of CITRUS , *i.e.*, $T$ (auto-regressive order), $emb$ (dimension of embedding size in the spatiotemporal encoder), $hid$ (dimension of linear mapping size in the spatiotemporal encoder), $F_{\text{MLP}}$ (dimension of linear mapping size in the MLP layers), $n_{\text{CITRUS}}$ (number of CITRUS blocks), $F$ (second dimension of $\mathbf{W}_l$ in (6)), $k_i$ (number of selected eigenvalue-eigenvector pairs in the $i$-th factor graph), $lr$ (learning rate), $n_{\text{batch}}$ (batch size), and $n_{\text{epochs}}$ (number of epochs), are provided in Table 9. Full details can be found in the implementation codes in https://github.com/ArefEinizade2/CITRUS.

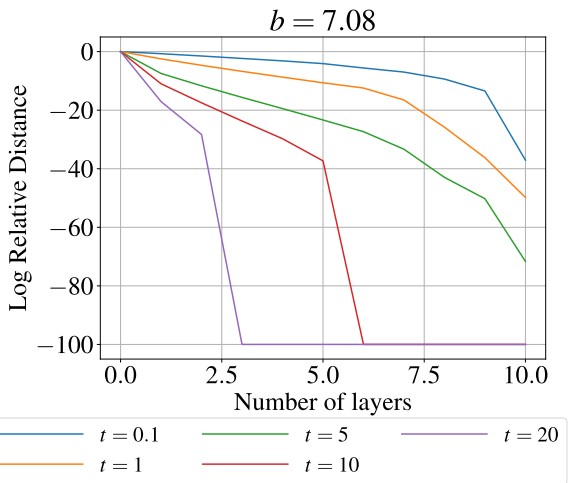

Figure 5: Log relative distance vs. increasing the number of layers for different values of $t$ in actual bounds.

## J   Experiments Compute Resources

All the experiments were run on one GTX A100 GPU device with 40 GB of RAM.

## K   Broader Impacts

The demonstrated superior performance of CITRUS in spatiotemporal forecasting tasks, such as traffic and weather prediction, can significantly benefit urban planning and public safety. Accurate traffic predictions can lead to better traffic management and reduced congestion, while improved weather forecasting can lead to serious urban inconvenience. This research can contribute to sustainable development goals by improving forecasting capabilities and decision-making processes, including sustainable cities and communities, climate action, and industry innovation and infrastructure.

