# OpenReview forum: "Continuous Product Graph Neural Networks"
_NeurIPS.cc/2024/Conference — NeurIPS 2024 poster_

### Official Review · Reviewer_Y9UP · 2024-07-11

**Soundness:** 2
**Presentation:** 3
**Contribution:** 2
**Rating:** 5
**Confidence:** 4

**Summary:**

The authors propose a new spectral GNN for Cartesian product graphs. The graph filter is chosen as Laplacian heat diffusion, which is separable across factor graphs. The stability and over-smoothing of the proposed model are studied. The authors perform synthetic experiments to validate the analysis. Experiments on real datasets also show the advantages of the proposed methods.

**Strengths:**

The authors provide a rather thorough analysis of the proposed methods. Theoretical results are provided. Experiments are detailed. The paper is well-written and easy to follow.

**Weaknesses:**

My main concern is with the over-smoothing analysis. The normalized product Laplacian as defined in Lemma 3.8. is generally not the normalized Laplacian of the product graph. It is more of a manufacture, so that the smoothness of the product graph decomposes to factor-wise smoothness terms. This also means that Eq. 15 is not true, if the LHS is defined as the 'actual' smoothness on the normalized product graph as in Eq. 13. I think the authors need to revise the analysis here.

Another concern is that the advantages of a separable graph filter are not made explicit. This is somewhat discussed in Remark 3.5., where the authors argue that factor-wise EVD is more efficient than EVD of the product graph. However, note that this is merely a consequence of the KP eigenvector structure of the Cartesian graph product [1], not the separability of the heat diffusion filter. You don't need a full EVD of the product Laplacian even if the filter is not separable. How a separable filter further benefits the computation is unclear to me.

Minor points: figure 2 is not very intuitive; figure 3 is also not very convincing.

[1] Stanley, Jay S., Eric C. Chi, and Gal Mishne. "Multiway graph signal processing on tensors: Integrative analysis of irregular geometries." IEEE signal processing magazine 37.6 (2020): 160-173.

**Questions:**

1. Spectral GNNs that use polynomials of graph adjacency matrices only require local aggregation and do not need to compute global EVD. How does the complexity of the proposed method compare to that? Are there good reasons for choosing Laplacian-based filters over adjacency-based filters in spectral GNNs?

2. Can the authors provide more details on the competing methods for real graph data? What about the methods that also explicitly consider the product structure? The real datasets all have a temporal dimension. How do traditional SP methods such as [2] perform?

[2] Grassi, Francesco, et al. "A time-vertex signal processing framework: Scalable processing and meaningful representations for time-series on graphs." IEEE Transactions on Signal Processing 66.3 (2017): 817-829.

**Limitations:**

Future directions to improve the theoretical results are discussed.

---

> ### Author Rebuttal · Authors · 2024-08-05
>
> **We’ve expanded some points of our rebuttal in additional comments. We kindly invite the reviewer to read them if required.**
>
> **W1**. We should have mentioned that, first, we need to define the Dirichlet energy for the tensorial data, $D_T(.)$, as the summation of the factor-wise Dirichlet energies on the unfolded metricized forms of the tensorial data $D_T(x):=\frac{1}{P}\sum_{p=1}^{P}{tr(\underline{X}\_{(p)}^\top\hat{L}\_p\underline{X}\_{(p)})}$, where $x=vec(\underline{X})$ and $\underline{X}_{(p)}$ is the $p$-th mode matricization of $\underline{X}$. The main intuition for this definition of $D_T(.)$ is that multi-domain data might be smooth on one of the factor graphs and non-smooth on the other factor graphs. Therefore, we cannot consider such cases as “pure” smooth signals on the product graph. This intuition aligns with relevant works in this direction [1,9], where these previous studies defined Diricihelet energy on the combinatorial Laplacian of the product graphs as the summation of the factor-wise Diricihelet energies. Note that the over-smoothing analysis in our paper holds in this case too. One can easily show that $D_T(x)$ is equivalent to writing the total Dirichlet energy using the product Laplacian as defined in Lemma 3.8 as $D_T(x)=x^\top \hat{L} x$, where $x=vec(\underline{X})$.
>
> Once we have this definition for the Dirichlet energy, our definition of the Laplacian matrix in Eqn. (14) emerges naturally. This matrix has two favorable properties: i) it’s positive semidefinite, and ii) its spectrum lies in $[0,2]$ (as in regular normalized Laplacians) having a bounded spectrum for numerical instability. We agree with the reviewer that this Laplacian might not be generally a valid normalized Laplacian for the product graph and deserves further exploration in future work. We’ve corrected some sentences and added a remark regarding this discussion in the Appendix section of the revised manuscript.
>
> **W2**. Let’s consider $h(L)=e^{-L\_{1}}\otimes \sum_{i=1}^{K}{L^{i}\_2}$ as a separable graph polynomial filter. Since $h(L)=(V_1\otimes V_2)(e^{-\Lambda_1}\otimes\sum_{i=1}^{K}{\Lambda^i_2})(V_1\otimes V_2)^\top$, the KP structure of the eigenvectors holds but we are not facing a valid well-defined graph product. However, instead of computing direct EVD of $h$ with complexity $(N_1N_2)^3$, we can first compute the factor graphs EVD with complexity $N_1^3+N_2^3$, and then $V_1\otimes V_2$ and $e^{-\Lambda_1}\otimes\sum_{i=1}^{K}{\Lambda^i_2}$. Although we agree about the relationships between having a heat diffusion filter defined on a Cartesian Laplacian and KP of separable heat filters, please note that computational benefits come from the possibility of selective eig-eiv pairs. For example, consider a heat diffusion defined on a Kronecker product graph adjacency. In this case, we have $h(A)=e^{-A_1\otimes A_2}=(V_1\otimes V_2)(e^{-\Lambda_1\otimes\Lambda_2})(V_1\otimes V_2)^\top$, and since we cannot write it as $h_1(A_1)\otimes h_2(A_2)$, we cannot use a selective strategy. While, in the Cartesian products, we can write $e^{-L_1}\otimes e^{-L_2}$ and also we have $e^{-L_1}=\sum_{i=1}^{N_1}{e^{-\lambda^{(1)}_i} v^{(1)}_1 {v^{(1)}_1}^\top}$. Therefore, selecting the most important pairs (as a low-rank approximation problem) has a theoretical justification, which is not generally true for Kronecker graphs.
>
> **W3**. These minor points are answered in the comments.
>
> **Q1**.
> - It has been shown that computing the forward propagation in a spectral GNN using the Chebyshev polynomials is $\mathcal{O}(K|\mathcal{E}|F\_{in}F\_{out})$, with $K$, $|\mathcal{E}|$, $F\_{in}$, and $F\_{out}$ being graph filter order, number of edges of the product graph, input and output features, respectively [4]. For our product graph model, the number of edges can be expressed as $|\mathcal{E}|=\sum_{p=1}^{P}{(\prod_{i=1\ne p}^{P}{N\_p})|\mathcal{E}\_p|}$ [5]. Therefore, the complexity of our forward propagation in Eqn. (7) is $\mathcal{O}(K\_P+K\_PF\_l+K\_PNF\_l+K\_PF\_lF\_{l+1}+NK\_PF\_{l+1})$, where $K_P=\prod_{p=1}^{P}{K_p}$. Assuming often $K_P\ll N$, our complexity approximately takes the form of $\mathcal{O}(K_PN(F_l+F_{l+1}))$. So, we can say that the complexity of our forward function is linear in terms of the number of nodes of the product graph, and can be reduced by selecting the most important eig-eiv pairs in factor graphs $K_P$. Note that, we also need one preprocessing step of the factor graph’s EVD with a complexity $\mathcal{O}(N^2_pK_p)$, but it's only needed once.
>
> - The PDEs on graphs are defined using Laplacian and not adjacency matrices, so using the adjacency leads to a lack of theory.
>
> **Q2**.
> - The full details about the baselines are provided in Appendix B.
> - The most general method that relies on product graphs, to the best of our knowledge, is GTCNN which is already compared in our paper. However, GTCNN is limited to only two-factor graphs.
> - Please note that we have already compared with the traditional (graph) SP methods GP-VAR [6] and GP-VARMA [6]. Indeed, GP-VAR and GP-VARMA are built upon [7].
>
> References:
> -----
> [1] “Low-rank and smooth tensor recovery on Cartesian product graphs” IEEE International Conference on Sampling Theory and Applications, 2023
>
> [2] “Graph neural networks exponentially lose expressive power for node classification” ICLR, 2020
>
> [3] “A note on over-smoothing for graph neural networks” 2020
>
> [4] “Convolutional neural networks on graphs with fast localized spectral filtering” NeurIPS, 2016
>
> [5] “Learning product graphs underlying smooth graph signals” 2020
>
> [6] “Forecasting time series with Varma recursions on graphs” IEEE TSP, 2019
>
> [7] “A time-vertex signal processing framework: Scalable processing and meaningful representations for time-series on graphs” IEEE TSP, 2017
>
> [8] "The emerging field of signal processing on graphs" IEEE SPM 2013
>
> [9] “Product graph learning from multi-domain data with sparsity and rank constraints.” IEEE TSP, 2021

---

> ### Author Response · Authors · 2024-08-05
> **More details about the responses**
>
> **W1**.
> - As a simple toy example, consider a $4\times3$ matrix $U=[1,2,3;1,2,3;4,5,6;4,5,6]$, and also assume the factor graph as 4 and 3-node simple path graphs. So, $U$ is smooth on the 4-node graph but it is not the case with the other 3-node one, and, so, can not be considered a pure smooth signal on the product graph.
>
> **W2**.
> - For the sake of completing the discussion, consider we could write the resulting multi-way graph filter as $h(L)=h\_1(L_1)\otimes h\_2(L\_2)$, where $h\_{1(2)}()$ is a polynomial graph filter (with possible infinite order like heat graph filters). Then, one can use low-rank expansion $h\_1(L\_1)=\sum_{i=1}^{K\_1}{h\_1(\lambda\_i) v\_i v^T\_i}$. Therefore, the computation can be improved by choosing an appropriate small enough $K_1$.
>
> **W3**.
> - Figure 2 illustrates the effect of factor stabilities on the overall stability, as Theorem 3.7 states. By varying adjacency SNR for each factor (related to $\mathcal{O}(\epsilon_p)$), this figure validates the presence of factor stability on the overall performance. We've included a 3D plot in the **uploaded PDF** on the main rebuttal (Fig. R5) with a more intuitive view of the same experiment for better illustrating the interpretation of Theorem 3.7.
>
> - Regarding Figure 3, please first note that it is on a logarithmic scale, which we forgot to mention and has been corrected in the revised version of the manuscript. The left plot is associated with the case of $\ln{s}-\frac{2\tilde{t}\tilde{\lambda}}{P}<0$, which is prone to over-smoothing and converges to zero very fast (exponentially). That's why the difference between $\ln{(\frac{E(X_l)}{E(X_0)})}$ and $l(\ln{s}-\frac{2\tilde{t}\tilde{\lambda}}{P})$ is extremely small. For the right plot in Figure 3, we have that  $\ln{s}-\frac{2\tilde{t}\tilde{\lambda}}{P}>0$. Here, again the theorem is validated by the RHS bound in Eqn. (17). However, this bound is not tight and we leave further analysis as future work. We kindly refer the reviewer to [2] to get more familiarized with this kind of analysis and plots regarding over-smoothing in discrete GNNs.

---

> ### Comment · Reviewer_Y9UP · 2024-08-10
> **Further Responses**
>
> I thank the authors for their detailed reply.
>
> W1. Please make sure to revise the paragraph at line 173. Also the author should realize that [9] used combinatorial Laplacian and that's why the smoothness is naturally decomposable. Normalized Laplacian is a whole different story. The authors should also pay attention to Eq.13 and Eq.15, and make clear distinction between them. Also make sure you don't accidentally use $E(U_t)$ as the form in Eq.13 in the proof.
>
> W2. I think one certainly can make the filter 'selective' even without separability. For example, consider a low-pass filter $f_i$ on some factor $i$ and a non-separable filter $g$ on the product graph. Their composition $f_i \circ g$ is not separable either, but selective on factor $i$. To be more specific, you can compute the EVD of factor graphs, transform the data to spectral domain, apply a non-separable filter, then select a subset of frequencies for each factor. I don't see how this is not as efficient as using a separable filter like the heat diffusion.

---

> ### Author Response · Authors · 2024-08-13
>
> Thank you for engaging in the discussion and for your helpful and insightful comments.
>
>
> **W1**. We will revise all that's required to align with the correct conclusions from our discussion.
>
>
> We agree that for the *general* normalized Laplacian, the decomposability might not be true. However, please notice that for our definition in Eqn. (14) in our paper, the decomposability holds. For example, if we consider two (general, and not even graph-related) factor matrices, the following relationship is valid for any general matrices $A,B,C$ with consistent dimensions [1]:
> - $AX+XB=C \leftrightarrow (A\oplus B^\top)vec(X)=vec(C).$
>
>
> Now, by considering $x=vec(X)$ and using this relationship, one can write:
> - $vec(X)^\top\left[(A\oplus B^\top)vec(X)\right]=vec(X)^\top\left[vec(AX+XB)\right]=vec(X)^\top vec(AX)+vec(X)^\top vec(XB)=tr(X^\top AX) + tr(X^\top XB)=tr(X^\top AX) + tr(XBX^\top).$
>
>
> If $A=\hat{L}_1$ and $B=\hat{L}_2$, the proof is completed. Please note that our over-smoothing analysis is easily adaptable for general combinatorial Laplacian matrices as well.
>
>
> **W2**. We agree that even in the non-separable filters, one can select a graph frequency from the factor graphs. However, we don't find a strong theoretical justification for this. There might exist a logic in specific cases, like in [2], but this requires the general study of the product graph's spectrum. For example, consider the non-separable filters are $h(\lambda_1,\lambda_2)=e^{\lambda_1\lambda^2_2-\lambda^2_1\lambda_2}$, or even $h(\lambda_1,\lambda_2)=\cos{(\lambda_1\lambda_2)}$. In these cases, the importance of $\lambda_1$ on the spectrum of the product graphs is also tied with the importance of $\lambda_2$. Therefore, the most important eigenvalue in the first-factor graph for performing low-pass filtering is not necessarily the one with the highest importance in the product graph spectrum. This is because the importance of the factor graphs' spectra is hindered by the non-separable function. Please notice that the theoretical justification is clear in our formulation since, based on our response in the previous rebuttal stage, the selection procedure in our paper comes from factor-wise low-rank approximation subproblems.
>
>
> As a final remark, please note that this whole discussion about the separability boils down to just one remark in our paper, and not to one of our main contributions. Therefore, we can easily make the relevant modifications without an important repercussion in our main takeaways.
>
>
> References
> ----
> [1] "The matrix cookbook", 2008
>
> [2] "Learning Cartesian Product Graphs with Laplacian Constraints", AISTATS, 2024.

---

### Official Review · Reviewer_QFxx · 2024-07-11

**Soundness:** 3
**Presentation:** 3
**Contribution:** 2
**Rating:** 5
**Confidence:** 3

**Summary:**

This paper proposes tensor PDE for graph neural networks temporal graph prediction. The construction is seems to be good and theoretical analysis for oversmoothing and stability are provided. Experiments show good performance with the proposed method compared to several baseline methods.

**Strengths:**

1) improvements on prediction accuracy fro the proposed method compared to existing methods.
2) Overall well-analyzed methods with some theoretical support on the behavior of the proposed method

**Weaknesses:**

1) There is no clear strategy to overcome oversmoothing.
2) Comprehensive details of hyperparameters is required. The lack of nature of hyperparameters and their ranges leads to difficulties in understanding the overall computational efficiency of the method.

**Questions:**

1) what are the training times for the CITRUS and its comparison to baseline methods (or the most relevant baseline)?

2) Over-smoothing effect is provided with a theorem and experimental validation, however, is there any suggestion or improvement in the proposed methods to overcome the over-smoothing?

3) what are the hyperparameters of CITRUS?  how would you tune them? Do you hyparameter tune for eigenvector-eigenvalue pairs? Does some quantity such as rank is required for the tensor construction?

4) Is there a specific structure of graph such as homophily and hitherophily in temporal graph that the proposed method is in more favor of?

**Limitations:**

yes

---

> ### Author Rebuttal · Authors · 2024-08-05
>
> **Q1**. The training time comparison with the most relevant baseline, GTCNN, has been provided in the **uploaded PDF** on the main rebuttal (Tab. R2) for the NOAA and MetrLA datasets. We observe that our model requires less time per epoch to be trained.
>
> **Q2**.
> - Based on Eqn. (18), we can expect to slow down the over-smoothing phenomenon by keeping $\tilde{t}<\frac{P}{2\tilde{\lambda}}\ln{s}$. This explains why we should not increase the receptive field parameter $t$, especially when dealing with strongly connected graphs or deep GNNs. This finding is consistent with previous research on over-smoothing in discrete GNNs [1]. In the **uploaded PDF** on the main rebuttal (Fig. R3), we show the values of the learned receptive fields $t$ by CITRUS across different numbers of horizons. For this experiment, we have $b=7.08$, where $b=\frac{P}{2\tilde{\lambda}}\ln{s}$. We observe in the figure that CITRUS always learns $t<b$ to slow down the over-smoothing for all horizons.
> - To further exemplify the implications of our theoretical analysis of over-smoothing, we now consider the graph receptive field $t$ as a hyperparameter in CITRUS and design an experiment for different values of $t$. In this experiment, we vary $t\in\{0.1,1,5,10,20\}$ and monitor the convergence to the over-smoothing state. We include the results of this experiment in the **uploaded PDF** on the main rebuttal (Fig. R4). We observe that for values of $t$ higher than $b$, CITRUS converges faster to the over-smoothing state for a larger number of layers. In practice, alleviating over-smoothing might not be enough to achieve good performance. Therefore, for $t<b$, we might need to perform other kinds of analyses like over-squashing, which we leave for future work.
>
> **Q3**. CITRUS has several hyperparameters related to typical architectural and optimization choices in neural networks, such as the number of layers, learning rate, weight decay, etc.  We’ve included all hyperparameters in the Appendix of the revised manuscript. Apart from these typical hyperparameters, the number of selected eigenvalue-eigenvector pairs $k$ is unique to our framework.
>
> We can choose an appropriate $k$ using two methodologies: supervised and unsupervised. For the supervised option, we can use cross-validation, as we did in our manuscript (Table 4). For the unsupervised method, we can analyze the Laplacian matrices for a low-rank approximation task. For example, in the **uploaded PDF** on the main rebuttal (Fig. R2), we plot the explained variances of the principal components (i.e., $||v\lambda v^\top ||_F^2$ for $v$ and $\lambda$ being the eigenvectors and eigenvalues) on the MetrLA dataset. We observe a strong concentration of variance in a few components. We also observe that we can capture almost 80% of the explained variance by only relying on 50 eigenvalue-eigenvector pairs. In this case, the recommended range of $k$ is about 10-25% of the number of nodes.
>
> Choosing an appropriate $k$ is indeed related to an efficient rank approximation on the factor Laplacians.
>
> **Q4**. Our framework does not make any assumption about homophily or heterophily properties of the underlying graphs. We have reported the intra-homophily (spatial dimension) and inter-homophily (temporal dimension) of the MetrLA and PemsBay datasets in Tab. R3 in the **uploaded PDF** on the main rebuttal, using the proposed algorithms in [2,3]. We observe in Tab. R3 that these datasets have a mix of low and high measures, indicating a combination of homophily and heterophily behaviors. Therefore, our framework can efficiently handle both homophily and heterophily cases.
>
> References:
> -----
> [1] “Graph neural networks exponentially lose expressive power for node classification,” ICLR, 2020
>
> [2] “Greto: Remedying dynamic graph topology-task discordance via target homophily,” ICLR, 2023
>
> [3] “Graph neural networks for graphs with heterophily: A survey”, 2022

---

> > ### Comment · Reviewer_QFxx · 2024-08-13
> > **Response to authors**
> >
> > Thank you for the response.
> >
> > I think the listing of the hyper-parameters and their ranges are important to have in the revised version of the paper. Furthermore, I think more studies should be helpful on commonly used hetherophilic graph datasets in the revised paper.

---

> ### Author Response · Authors · 2024-08-13
>
> Thank you for your insightful comments and for engaging in the discussion.
> - As suggested, we will list the range and precise values of our hyperparameters in the Appendix.
> - Regarding the heterophilic datasets, we should first mention that measuring the homophily indices in spatiotemporal settings is more challenging than in regular node classification scenarios. In our case, we are facing two intra-graph (the spatial dimension) and inter-graph (the temporal domain) aspects. The study in [1] considers Metr-LA, PemsBay, and temperature datasets (similar to Molene and NOAA used in our paper) as highly heterophilic.
> To complement our rebuttal, we’ve studied the heterophilic measurements on the Molene and NOAA datasets, which we provide in the following table (${\pi}^{s}\_{v\_i}$ and ${\pi}^{T}\_{v\_i}$ corresponds to intra and inter-graph homophily, respectively). Due to the wide range of variability across these datasets, we have already evaluated our performance on various datasets in terms of homophily-heterophily levels. For instance, Molene is highly heterophilic in both the intra and inter-graph domains, while NOAA acts more homophilic than Molene in the inter-graph domain. We can also compare the provided metrics for NOAA with the Temperature or KnowAir datasets used in [1], where we observe similar dynamics. We will include these discussions related to homophily-heterophily metrics in the Appendix of the revised manuscript.
> | Datasest |  ${\pi}^{s}\_{v\_i}:p\_s$ | ${\pi}^{s}\_{v\_i}:q\_p$ | ${\pi}^{s}\_{v\_i}:q\_n$ | ${\pi}^{T}\_{v\_i}:p\_s$ |  ${\pi}^{T}\_{v\_i}:q\_p$ | ${\pi}^{T}\_{v\_i}:q\_n$ |
> |----------|----------|----------|----------|----------|----------|----------|
> | MetrLA   | 0.2273 | 0.4732 | 0.2995 |  0.3325 | 0.4920 | 0.1755 |
> | PemsBay    | 0.1073 | 0.5912 | 0.3015 |  0.2399 | 0.6863 | 0.0738 |
> | Molene    | 0.0148 | 0.6248 | 0.3602 | 0.0152 | 0.5545 | 0.4301 |
> | NOAA    | 0.1249 | 0.5345 | 0.3405 | 0.2862 | 0.4351 | 0.2786 |
>
> References:
> ---
> [1] "Greto: Remedying dynamic graph topology-task discordance via target homophily”, ICLR, 2023

---

### Official Review · Reviewer_gs8b · 2024-07-12

**Soundness:** 3
**Presentation:** 2
**Contribution:** 3
**Rating:** 6
**Confidence:** 3

**Summary:**

The paper proposes CITRUS, a novel model for jointly learning multidomain couplings from product graph signals based on tensorial PDEs on graphs (TPDEGs). By modelling these representations as separable continuous heat graph kernels as solutions to the TPDEG, the paper shows that the underlying graph is actually the Cartesian product of the domain-specific factor graphs. Then the paper studied the stability and over-smoothing aspects of CITRUS theoretically and experimentally. Finally, the paper applies CITRUS to tackle the traffic and weather spatiotemporal forecasting tasks on public datasets, illustrating its effectiveness compared to state-of-the-art methods.

**Strengths:**

1. The paper is overall well written and easy to follow (except some notations could be better defined).
2. Handling multi-domain graph data is an interesting and challenging problem in practice.
3. Learning joint (and not sequential) spatiotemporal couplings by modeling these dependencies using product graphs with learnable receptive fields is an interesting idea that is shown to be effective.
4. The evaluation is done with respect to a diverse set of baselines.

**Weaknesses:**

1. Bad notations: too many tilde, lower bar, subscript that seems unnecessary to me. They make it hard to read the math. Math is hard to communicate so good writing is essential.
- For example, what is purpose of having tilde in (3) and (4)? Then it disappeared in (5).
- Is lower bar for $U$ necessary? why not just go with $U$?
- is $\times_i$ mode-i tensorial multiplication? what is the precise definition?
- One minor thing --- $:=$ is used in line 120 but not line 123. Please check for consistency of other notations as well.
2. The bound in Theorem 3.9 seems to be too loose to be meaningful in some cases, as shown in Figure 3, right plot that while the bound suggests divergence, the actual $E(X_l)$ converges to zero. Also, what is the implication of Theorem 3.9 on over-smoothing besides that we can focus on one factor graph for the phenomenon?

**Questions:**

1. Could the authors provide the standard deviation for the experimental results reported in table 1-4?
2. What are typical problems involving more than two factor graphs?
3. I see that Cartesian product arises in solutions to the TPDEG, that is why the CITRUS uses Cartesian product. I wonder if the authors have any intuition why this product is a good for modelling spatiotemporal couplings in practice.
4. It is also interesting that CITRUS seems to outperform especially in longer horizon predictions. Any investigation why it is the case?
5. Any guided way to choose k? Also any idea why a very small k seems to be a good enough approximation?

---

> ### Author Rebuttal · Authors · 2024-08-01
>
> **W1**. We’ve significantly simplified the notations in our paper. For example, we removed the unnecessary tildes from the notation in the revised manuscript.
> - Regarding the lower bar for tensors, we find it important to differentiate tensors (higher-order data) from regular matrices as in [9], but, if you have found it hard to read, we will think of alternating notations, like in [1].
> - For the tensor product, this is an extension of the matrix multiplication operation. For example, on 3D tensors, $\underline{X}=\underline{G}\times\_1 A$ is equal to $\underline{X}\_{(1)}=A\underline{G}\_{(1)}$, where the matrix $\underline{G}_{(1)}$ is obtained by concatenating the mode-1 slices of the tensor $\underline{G}$. We’ve included a detailed description in the Appendix of the revised manuscript. For more information please refer to Sections 2.4-5 in [1].
> - In line 120, we have the definition of the product Laplacian, but the equations in line 123 are EVD forms of Laplacians, not definitions themselves.
>
> **W2**.
> - In Fig. 3, we forgot to mention that the y-axis is on a logarithmic scale. Indeed, both lines are very close in non-logarithmic scales. We’ve corrected the label on the y-axis. Regarding the divergence, please note that our theorem states that $\ln{(\frac{E(X_l)}{E(X_0)})}\le l(\ln{s}-\frac{2}{P}t\lambda)$. Therefore, when $\ln{s}-\frac{2}{P}t\lambda$ is positive, the RHS here grows with an increase in the number of layers. On the other hand, there is no problem with $E(X_l)$ going to zero because the theorem holds and is validated by the experimental results. However, in this case, the bound is looser than the case of $\ln{s}-\frac{2}{P}t\lambda<0$.
> - Theorem 3.9 aims to characterize and provide insights about the over-smoothing aspects of CITRUS. To see the effect of all factor graphs, by using Lemmas A.2-A.4, we can also obtain $E(X_{l})\le e^{l\left(\ln{s}-\frac{2\sum_{p=1}^{P}{t^{(p)}\lambda^{(p)}}}{P}\right)}E(X_{0})$. Therefore, we observe that over-smoothing is affected by the weighted average of the factor graph spectra, weighted by their receptive fields. We have provided further over-smoothing diagrams for varying sparsity levels for factor ER graphs in the **uploaded PDF** on the main rebuttal (Fig. R1). We observe that the denser the factor graphs (i.e., higher values for edge probability $p$), the faster the convergence to the overall over-smoothing state. This addresses the reviewer’s concern about the factor graph effects on over-smoothing. This aligns with results in the literature, e.g., [2], about a higher possibility of over-smoothing with denser graphs. We’ve added this discussion as a Remark in the Appendix of the revised manuscript.
>
> **Q1**. The standard deviations (STDs) have been provided in the **uploaded PDF** (Tab. R1) on the main rebuttal. Additionally, please note that we included some STDs in Tables 5, 6, and 7 in the Appendix.
>
> **Q2**. There are several possible examples of product graphs with more than two-factor graphs. For example, video data can be represented as a three-factor graph (width, height, and time). Similarly, in sleep staging with brain signals, there are multiple dimensions, such as temporal sleep windows, the spatial dimension of electrodes on the head, the frequency domain, and the time domain [3].
>
> Another possible application could be in recommendation systems, where we have an item graph, a user graph, and a feature space of the user-item elements.
>
> **Q3**. Cartesian product graphs are useful for processing spatiotemporal data because we can model it as the Cartesian product of a spatial graph and a simple path graph (time dimension).  By copying the spatial graph through time, one can model time-varying graph signals. This type of modeling has a rich literature in graph signal processing. The reviewer is kindly referred to [4-7], and also to Section 5 in the first chapter of [8].
>
> **Q4**. The main difference between CITRUS and other baselines lies in its ability to learn the graph receptive fields $t$, which is difficult for other baselines (with fixed or non-adaptive receptive fields) to estimate. We've plotted the learned $t$ for different numbers of horizons in the **uploaded PDF** on the main rebuttal (Fig. R3). We observe that the estimation of $t$ is more robust for longer horizons since $t<b=P\ln{s}/2\lambda$ (here, $b=7.08$). According to our theoretical analysis, this alleviates the over-smoothing phenomenon for longer horizons.
>
> **Q5**. We can choose an appropriate $k$ using two methodologies: supervised and unsupervised. For the supervised option, we can use cross-validation, as we did in our manuscript. For the unsupervised method, we can analyze the Laplacian matrices for a low-rank approximation task. For example, in the **uploaded PDF** on the main rebuttal (Fig. R2), we plot the explained variances of the principal components of the spatial Laplacian (i.e., $||v\lambda v^\top ||_F^2$ for $v$ and $\lambda$ being the eigenvectors and eigenvalues) on the MetrLA dataset. We observe a strong concentration of variance in a few eigenvalues. We also observe that we can capture almost 80% of the explained variance by only relying on 50 components.
>
> References:
> -----
> [1] “Tensor decompositions and applications,” SIAM Review, 2009
>
> [2] “Graph neural networks exponentially lose expressive power for node classification,” ICLR, 2020
>
> [3] “Learning product graphs from spectral templates,” IEEE TSIPN, 2023
>
> [4] “Big data analysis with signal processing on graphs”, IEEE SPM, 2014
>
> [5] "Product graph learning from multi-domain data with sparsity and rank constraints,” IEEE TSP, 2021
>
> [6] "Learning product graphs from multidomain signals,” IEEE ICASSP 2020
>
> [7] “Product graph Gaussian processes for multi-domain data imputation and active learning,” IEEE EUSIPCO, 2023
>
> [8] "Vertex-frequency analysis of graph signals", Springer, 2019
>
> [9] "Era of big data processing: A new approach via tensor networks and tensor decompositions". 2014

---

> > ### Comment · Reviewer_gs8b · 2024-08-10
> > **Response to author's rebuttal**
> >
> > Thank the authors for their rebuttal. There are a few further comments and questions I want to discuss:
> >
> > W2: I guess my concern was not that the Theorem is wrong but not tight enough. It seems that the authors admit that the current bound might be vacuous in certain cases. Is there a way to make the results tighter?
> >
> > Q4: What do you mean when saying "the estimation of  is more robust for longer horizons" in Fig R3 --- do you mean the variance is smaller? I can see that this learned t is not under the scenario of oversmoothing established in the theoretical results and hence it helps alleviate oversmoothing. But I wonder if alleviating oversmoothing itself is the only reason behind the good long horizon performance (so other methods are bad because they oversmooth?)

---

> ### Author Response · Authors · 2024-08-13
>
> Thank you for engaging in the discussion.
>
>
> **W2**. While it's true the right plot of Fig. 3 is not tight enough, we should emphasize this is an open research question. Our findings align with previous work in over-smoothing, please refer to Fig. 2 in [1] for example, where a similar loose bound was found.
>
>
> We now provide insights for possible future directions to find a tighter bound:
> - The RHS in Theorem 3.9 also depends on the maximum singular value of the learnable weights matrices $W\_{l1},...,W\_{ll}$, *i.e.*, $s$, which solely depends on the training process. We can use techniques like weight normalization [1] to bound $s$.
> - When there's a big gap between the arguments of the Lipschitz functions, we might use integral Lipschitz functions [2] since in that case $|f(x\_2)-f(x\_1)|\le C\frac{|x\_2-x\_1|}{|x\_2+x\_1|/2}$. This is tighter than an absolute Lipschitz function, but its effect on the factor graph spectrums needs further theoretical exploration.
> - Previous works have found that the Frobenius norm might not be very tight as an upper bound for the spectral norm [3]. Precisely, $\\|T\\|\le\\|T^K\\|^{\frac{1}{K}}=\sup\_{x\ne 0}{\left(\frac{\\|T^Kx\\|}{\\|x\\|}\right)^{\frac{1}{K}}}=\sup\_{x:\\|x\\|=1}{(\\|T^Kx\\|)^{\frac{1}{K}}}$, for some finite $K$ [3]. Since $E(x)=x^\top L x=\\|L^{1/2}x\\|\_F^2=\sum_{i=1}^{N}{\lambda\_i\tilde{x}^2\_i}$ with $\tilde{x}$ being the GFT of $x$ (which is a weighted nuclear norm with direct relationship with the spectral norm), we might use other norms like Sobolev [4], gradient-adapted [5], or other $p-$norms [4,6].
> - Recently, some works have explored alternatives to the regular triangular inequalities [7], which might not be tight for the over-smoothing analysis.
>
>
> In summary, there are some future directions to make our bounds tighter, but they need to be carefully studied in the case of continuous filters in product graphs.
>
>
> **Q4**. By robustness, we mean smaller standard deviations in prediction performance as shown in Tab. R1 in the **uploaded PDF file of the main rebuttal**.
>
>
> Regarding the longer horizons, the authors from the GTCNN model [8] (which can be considered as a particular case of CITRUS for discrete, two-factor graphs) said: "The GTCNN outperforms the other models in a short horizon while Graph WaveNet and GMAN work better for longer horizons. The benefits in the short term are due to high-order spatiotemporal aggregation in the GTCNN which allows capturing efficiently the spatiotemporal patterns in the data". Therefore, we state the GTCNN model doesn't perform well in longer horizons due to the non-optimal receptive field of the discrete case. More formally, consider a one-layer CITRUS network without non-linearity for predicting one horizon as $\tilde{y}=e^{-tL\_{\diamond}}Xw$. The loss function is therefore given by $\min\_{w}{\\|y-e^{-tL\_{\diamond}}Xw\\|\_2^2}=\min\_{w}{w^\top X^\top e^{-2tL\_{\diamond}}Xw-2y^\top e^{-tL\_{\diamond}}Xw}$. Here, the second term tries to maximize the inner product (the similarity) between the target $y$ and predicted output $\tilde{y}=e^{-tL\_{\diamond}}Xw$. The first term $f_{CITRUS}=w^\top X^\top e^{-2tL\_{\diamond}}Xw$ enforces the output to be as smooth as possible on the heat diffusion filters $e^{-tL\_{\diamond}}$. For a simple GTCNN [8], this smoothness term takes the form of $f\_{GTCNN}=w^\top X^\top L\_{\diamond}^2Xw$. If we consider longer horizons, these terms will accumulate leading to bigger smoothness terms in the loss function, likely leading to over-smoothing. The key difference between $f\_{CITRUS}$ and $f\_{GTCNN}$ is that our model learns $t$ to alleviate this accumulation issue as observed in Fig. R3 in the **uploaded PDF file of the main rebuttal**. Please also note that we can also have better control of the spectrum of $f\_{CITRUS}$ because of the learnable heat kernel.
>
>
> Apart from the role of $t$ in $f_{CITRUS}$, other aspects besides over-smoothing could also play an important role, like over-squashing and over-fitting, which we leave for future work.
>
>
> References
> ----
> [1] “Graph neural networks exponentially lose expressive power for node classification”, ICLR, 2020
>
> [2] “Stability properties of graph neural networks”, IEEE TSP, 2020
>
> [3] “Learning interface conditions in domain decomposition solvers”, NeurIPS, 2022
>
> [4] “Reconstruction of time-varying graph signals via sobolev smoothness”, IEEE TSIPN, 2022
>
> [5] “Time-varying graph signal reconstruction”, IEEE JSTSP, 2017
>
> [6] “Flow smoothing and denoising: Graph signal processing in the edge-space”, IEEE GlobalSIP, 2018
>
> [7] “Studying the effect of gnn spatial convolutions on the embedding space’s geometry”, UAI, 2023
>
> [8] "Graph-time convolutional neural networks: Architecture and theoretical analysis", IEEE TPAMI, 2023.

---

> ### Comment · Reviewer_gs8b · 2024-08-13
> **Thank you**
>
> I thank the authors for their detailed answers, which help me understand the work better. For Q4, one thing I am not sure is that whether the smaller standard deviations in prediction performance shown in Tab. R1 are due to more accurate estimation of $t$ in longer horizons --- as the trend for standard deviations seems similar for GTCNN.
>
> I will keep my score and stay on the positive side.

---

> ### Author Response · Authors · 2024-08-13
>
> We thank you again for the insightful discussion.
>
> Regarding your mentioned point, actually, this is almost what we expected based on our previously provided detailed response, in which we showed both GTCNN and CITRUS naturally embed a graph filter smoothness term on the product graphs in their loss functions. Based on the deep study of the behavior of the smoothness regularization terms [1,2,3], they essentially do not allow the response to have high variations (around the mean). But, as we already outlined in our previous response, this term has a pure accumulating nature in GTCNN leading to (most probably) higher importance to gain a smooth response, while, in CITRUS, this term is getting adapted and modified with learnable importance (due to learnable graph receptive fields in our approach). In summary, we expected smooth results from both GTCNN and CITRUS, but more accurate and optimized for CITRUS.
>
> Thanks again.
>
> References:
> ---
> [1] "Graph regularized nonnegative matrix factorization for data representation." IEEE TPAMI, 2010
>
> [2] "Fast robust PCA on graphs." IEEE JSTSP, 2016
>
> [3] "Robust principal component analysis on graphs." ICCV, 2015

---

### Official Review · Reviewer_wQ9d · 2024-07-13

**Soundness:** 3
**Presentation:** 3
**Contribution:** 3
**Rating:** 6
**Confidence:** 2

**Summary:**

The authors propose Tensorial Partial Differential Equations (TPDEG) to model multidomain data. Then they propose Continuous Product Graph Neural Networks (CITRUS) as a continuous solution. They provide theoretical and experimental analysis of their proposed approaches. They test their approach on spatiotemporal forecasting tasks and show SOTA performance.

**Strengths:**

* The paper is well written.
* The proposed methods are backed by theory.
* The experimental results are SOTA.
* Ablation study is provided.

**Weaknesses:**

See questions.

**Questions:**

1. Is it possible that real world graphs won't be representable as a cartesian product of factor graphs? How much approximation error is observed due to this assumption?

**Limitations:**

The authors address the limitations of their work as the last paragraph of their paper.

---

> ### Author Rebuttal · Authors · 2024-07-31
>
> One example where real-world graphs cannot be represented as a Cartesian product of factor graphs is the case where we have time-varying dynamic graphs, which might lead to some approximation errors by assuming the resulting graph is a Cartesian product graph. Indeed, this comment relates to the general issue of inaccurate adjacencies, denoted as $\tilde{A}=A+E$, which we studied in the stability analysis of CITRUS in Section 3.2. Therefore, the approximation error is bounded in Theorem 3.7.
>
> Another case is when the real-world graphs follow a different product graph operation. For example, let's consider two well-known graph products: the Strong product and the Kronecker product. Suppose the true graph is actually a Strong product graph, $A=A_1\otimes A_2+A_1\otimes I_2+I_1\otimes A_2$, but it is mistakenly treated as a Cartesian product $\tilde{A}=A_1\otimes I_2+I_1\otimes A_2$. In this case, the error $E$ is given by $E=A-\tilde{A}=A_1\otimes A_2$, where $||E||=\lambda^{(1)}\_{max}\lambda^{(2)}\_{max}$. Therefore, the approximation error depends on the multiplication of the maximum eigenvalues of the factor graphs.
>
> Similarly, if we consider Kronecker product graphs, $\tilde{A}=A_1\otimes A_2$, we find that $||E||\le\lambda^{(1)}\_{max}+\lambda^{(2)}\_{max}+\lambda^{(1)}\_{max}\lambda^{(2)}\_{max}$.  In this case, the summation of the maximum eigenvalues of the factor graphs matters too.
>
> In summary, the bound on the approximation error depends on the spectrum of the factor graphs.

---

> > ### Comment · Reviewer_wQ9d · 2024-08-12
> >
> > Thank you for reply. I acknowledge reading the rebuttal.

---

### Author Rebuttal · Authors · 2024-08-05

We express our gratitude to the reviewers for their thoughtful and constructive feedback. We are encouraged by their recognition of several strengths in our work. In particular, the reviewers found that: "*The proposed methods are backed by theory*" (reviewers wQ9d, QFxx, and Y9UP), "*The experimental results are SOTA*" (reviewers wQ9d and QFxx), "*The paper is overall well written and easy to follow*" (reviewers wQ9d, gs8b, and Y9UP), and the paper presents a "*thorough analysis of the proposed methods*" (reviewers QFxx and Y9UP). Besides, they found that “*handling multi-domain graph data is an interesting and challenging problem in practice*” (reviewer gs8b).

We have addressed each of the reviewers' comments in a detailed manner, focusing on clarifying any points of uncertainty and resolving any misunderstandings that may have arisen. Our responses are organized in a point-by-point format, as outlined in the subsequent sections of this rebuttal. A summary of these responses is listed as follows:

* We have provided additional analyses on the over-smoothing phenomenon and discussed how to alleviate it.

* We discussed the approximation error that occurs due to the Cartesian product assumption when a graph cannot be fully represented as a Cartesian product of factor graphs.

* We clarified the methodology used for tuning and analyzing the hyperparameter of CITRUS.

* We added additional theoretical and experimental comparisons with the most relevant baselines and spectral GNNs to gain more insights into the advantages of our method.

* We clarified the main intuition behind using Cartesian product graphs to model time-varying spatiotemporal time series.

* We outlined the ability of the proposed method to handle both homophily and heterophily in graphs.

* We highlighted the computational advantages of the separability of the graph filters.

**We have uploaded a one-page PDF file and included additional results as tables and figures. We kindly invite reviewers to refer to them whenever required.**

---

### Decision · Program_Chairs · 2024-09-25

**Decision:**

Accept (poster)

**Comment:**

Overall, the reviewers agree that this is a good publication in terms of the proposed approach, clarity, theoretical analysis, and experimental results, and based on these merits, the paper should be accepted for publication. The feedback provided to the authors by the reviews as well as the additional experiments the authors conducted in the rebuttal should be included to improve the final version of their paper.